# ReGRAF: Training free Prompt Refinement via Gradient Flow for Segmentation

## Abstract

Visual Foundation Models (VFMs) such as the Segment Anything Model (SAM) have significantly advanced segmentation tasks. However, SAM and its variants necessitate substantial manual effort for prompt generation and additional training for specific applications. Recent approaches have addressed these limitations by integrating SAM into one-shot and few-shot segmentation, enabling auto-prompting through semantic alignment between query and support images. Despite these advancements, they still generate inadequate prompts that degrade segmentation quality due to visual inconsistencies between support and query images. To tackle this limitation, we introduce ReGRAF (**Re**finement via **Gra**dient **F**low), a training-free method that refines prompts through gradient flow derived from SAM's mask decoder. ReGRAF easily integrates into auto-prompting segmentation frameworks and is theoretically proven to refine segmentation masks with high efficiency and precision. Extensive evaluations demonstrate that ReGRAF consistently improves segmentation quality across various benchmarks, effectively mitigating inadequate prompts without requiring additional training or architectural modifications.

## 1 Introduction

Large models have paved a new era of foundation models, e.g., Large Language Models (LLMs) (Zeng et al., 2022; Touvron et al., 2023; Brown et al., 2020), demonstrating exceptional versatility and generality across various tasks. In computer vision, Visual Foundation Models (VFMs) have emerged to revolutionize vision tasks such as image classification (Dosovitskiy et al., 2021; Touvron et al., 2021), object detection (Carion et al., 2020; Liu et al., 2021) and segmentation (Cheng et al., 2021; Strudel et al., 2021) by leveraging advanced architectures and enormous data. Specifically in segmentation, Segment Anything Model (SAM) (Kirillov et al., 2023) arises as a universal VFM which addresses challenges of segmenting objects with diverse appearance by taking prompts such as points, bounding boxes, and coarse masks.

While SAM is versatile, it often struggles with coarse mask boundaries and small holes scattered throughout the segmentation mask in complex situations. (e.g. segmenting multiple objects or small parts of an object.) This limitation has led to utilizing fine-tuning methods aimed at improving its precision (Chen et al., 2023; Ke et al., 2024; Wu et al., 2023). These approaches introduce additional optimization for SAM with different segmentation scenarios to handle diverse object appearances and complexities. However, notice that SAM and its fine-tuned variants *are not automated*; they require label-intensive efforts to identify "proper prompts" for segmenting objects of interest. Additionally, these fine-tuned variants necessitate extra learnable parameters and training on additional datasets.

Recent works address the issues above by integrating SAM and its variants into one-shot and few-shot segmentation scheme, enabling auto-prompting by leveraging the semantic alignment between a query and support images. For instance, PerSAM (Zhang et al., 2024) relies on one-shot data, comprising a support image and a rough mask of the target object. It utilizes a similarity map between a query and target object embeddings to locate the target object precisely to facilitate prompt generation. Further more, PerSAM-F, a fine-tuning variant of PerSAM, introduces learnable parameters to fine-tune SAM, producing a linear combination of potential segmentation masks of different hierarchical levels to resolve ambiguity (Zhang et al., 2024). Similarly, Matcher effectively combines pre-existing VFMs to tackle one/few-shot segmentation tasks without additional training (Liu et al., 2024).

| (a) Support Image | (b) PerSAM-F | (c) Matcher | (d) ReGRAF | (e) Ground Truth |

Figure 1: **Example of prompts (green dots) and segmentation results from VFM.** (a) Support Image (elephant), (b) and (c) Misaligned prompts and segmentation results from PerSAM and Matcher, (d) Result from Matcher refined by ReGRAF, (e) Ground Truth.

While the aforementioned methods have been successful, variations in appearance (e.g., color, viewpoint, and shape) between query and support images often lead to misleading similarity maps. This causes auto-prompting segmentation frameworks (e.g. PerSAM-F and Matcher) to generate inadequate prompts (false-positive, semantically ambiguous, or semantically insufficient prompts) that confuse SAM's decoder in capturing the target object(s), thereby compromising the quality of the segmentation mask. For example, Fig. 1b and Fig. 1c illustrate instances where PerSAM-F and Matcher generate semantically insufficient prompts, failing to fully capture the entire target object (e.g., the elephant) within the scene. In such a scenario, we hypothesized that the gradient flow from the mask decoder encapsulates comprehensive contextual information learned during SAM's pretraining. By iteratively refining the initial prompts based on this information, we expect obtaining higher-quality segmentation masks, as illustrated in Fig. 1d.

To this end, we propose a framework **Re**finement via **GRA**dient **F**low (**ReGRAF**) of mask decoder which minimizes the entropy-regularized Kullback-Leibler (KL) divergence between the query and support embeddings through the gradient flow derived from the mask decoder. This flow aligns the query embedding with the semantics of the target object, improving the similarity map and eventually enhancing the quality of segmentation masks. Additionally, we provide a rigorous theoretical analysis that guarantees the convergence of this process. Specifically, we prove that the gradient flow converges exponentially to the optimal probability density function (pdf) of the VFM's embedding space, ensuring robust refinement with a limited number of iterations. As the existing methods significantly rely on similarity, our method improves upon existing methods that are grounded on auto-prompting segmentation frameworks. In summary, our key contributions are as follows:

- **Simple and Effective Training-Free Refinement Method:** We introduce a novel, training-free refinement method that improves the segmentation mask quality of promptable segmentation models, without additional learnable parameters or datasets.

- **Broad Applicability to Auto-Prompting Segmentation Frameworks:** Our method offers wide adaptability and can be easily integrated with any existing segmentation framework that utilizes auto-prompting techniques.

- **Theoretical Guarantee of Convergence:** We provide a convergence analysis demonstrating that ReGRAF drives query embeddings to converge exponentially to the optimal embedding distribution for segmentation masks.

We empirically show the effect of ReGRAF through qualitative and quantitative experimental results. ReGRAF is validated on five independent datasets with two different tasks (i.e., semantic and part segmentations), which demonstrate improvements as we initially hypothesized.

## 2 RELATED WORKS

### 2.1 ONE/FEW-SHOT SEGMENTATION

Few-shot segmentation aims to develop models capable of segmenting new classes using only a few annotated samples (e.g., one to five) (Shaban et al., 2017), making it particularly useful in scenarios where acquiring large amounts of annotated data is difficult. Typically, these models use a pre-trained feature extractor to process both support and query images, combining information from support images to perform segmentation (Zhang et al., 2022; Hong et al., 2022).

Recently, several approaches leveraging SAM as a pre-trained model have been proposed. PerSAM relies solely on one-shot data, comprising a support image and a rough mask of the target object. It utilizes a similarity map between a query and target object embeddings to locate the target object precisely, facilitating prompt generation. However, PerSAM faces challenges in cases of visual ambiguity, such as objects with visually distinct subparts or hierarchical structures (Zhang et al., 2024).

PerSAM-F, fine-tuning variant of PerSAM, addresses this problem by introducing two learnable parameters to fine-tune SAM, producing a linear combination of potential segmentation masks of different hierarchical levels to resolve ambiguity (Zhang et al., 2024). This approach improves segmentation accuracy and prevents overfitting on one-shot data, demonstrating enhanced performance in challenging scenarios.

Similarly, Matcher (Liu et al., 2024), effectively combines pre-existing VFMs to tackle one/few-shot segmentation tasks without additional training. Matcher utilizes DINOv2 (Oquab et al., 2024) as an encoder to compute a similarity map with semantic understanding to accurately identify the location of a target object in the query image. It then employs SAM as a segmenter to obtain segmentation masks, benefiting from the zero-shot segmentation performance of the SAM.

## 2.2 FINE TUNING VARIANTS OF SAM

For efficient adaptation of SAM to various downstream tasks, a range of fine-tuning variants has emerged. HQ-SAM, for instance, adds a high-quality output token and trains on the HQSeg-44K dataset to improve mask precision (Ke et al., 2024). VRP-SAM (Sun et al., 2024) implements a visual reference prompt encoder, requiring specific reference images and detailed training. SAM-Adapter (Chen et al., 2023) modifies the architecture by adding lightweight adapter layers while keeping most of SAM's original parameters frozen, reducing training effort. MobileSAM (Zhang et al., 2023) replaces the heavy ViT-H encoder with Tiny-ViT, reducing the model size and complexity while still requiring adaptation.

All of these methods, however, necessitate additional training data and structural changes to the model architecture. These fine-tuning strategies still heavily depend on manual prompts, with training datasets requiring carefully crafted prompt annotations alongside segmentation data. This reliance on curated prompts limits their applicability in real-world settings, where obtaining accurate prompts and consistent user interaction may be challenging.

## 2.3 PROMPT TUNING

Prompt tuning has emerged as a lightweight and efficient fine-tuning strategy, enabling models to leverage their extensive pre-trained knowledge without the need to significantly modify their internal parameters (Lester et al., 2021; Sun et al., 2023).

In computer vision, especially with vision foundation models like SAM, prompt tuning can be thought of as modifying the visual clues that are provided to guide the model in segmentation (e.g. points, bounding boxes, or masks). Instead of retraining SAM for new segmentation tasks, prompt tuning focuses on refining these input prompts to better capture object boundaries and features.

Recent works like PerSAM-F and Matcher are specialized forms of prompt tuning, refining the input prompts or embeddings for segmentation tasks. However, these methods primarily rely on similarity between query and support images, which can lead to inadequate prompts and sub-optimal segmentation performance when the images have appearance variations.

## 3 METHOD

This section describes the overall process of ReGRAF. We first introduce the key concepts of the gradient flow of entropy-regularized KL-divergences, followed by a detailed explanation of refinement via gradient flow of segmentation mask decoder.

### 3.1 GRADIENT FLOW OF ENTROPY-REGULARIZED KL-DIVERGENCES

This subsection briefly describes the construction of the gradient flow of entropy-regularized KL-divergences; more details can be found in the overview in Santambrogio (2017) and discriminator gradient flow (Ansari et al., 2021) that forms the basis of our research.

Let $\mathbf{v}$ be a vector of interest (e.g. a latent vector of a query image), and let $\rho(\mathbf{v})$ and $\mu(\mathbf{v})$ be the candidate and target (ideal) pdfs over $\mathbf{v}$, For theoretical simplicity, we restrict our discussion to pdfs defined on the latent vector space $\mathbf{v}$ that belong to the 2-Wasserstein space $\mathcal{W}_2$.

We now employ the gradient flow to update $\mathbf{v}$ in order to make the candidate distribution $\rho$ approximate the target distribution $\mu$. Specifically, our focus lies in minimizing the entropy-regularized KL-divergence $\mathrm{F}_\mu(\rho)$ between the target distribution $\mu$ and its candidate distribution $\rho$ as:

$$
\begin{aligned}
\min_\rho \mathrm{F}_\mu(\rho) &= \min_\rho \left\{ \mathrm{KL}(\mu\|\rho) - \gamma \mathrm{H}(\rho) \right\} \\
&= \min_{\mathbf{v}} \left\{ -\int \log(\rho(\mathbf{v})/\mu(\mathbf{v}))\rho(\mathbf{v})dx + \gamma \int \rho(\mathbf{v}) \log(\rho(\mathbf{v}))d\mathbf{v} \right\},
\end{aligned}
\tag{1}
$$

where $\mathrm{KL}(\mu\|\rho)$ denotes the KL-divergence between two distributions $\mu$ and $\rho$, $\mathrm{H}(\cdot)$ denotes the entropy function, and $\gamma > 0$ is a hyperparmeter that controls the strength of the entropy regularization. Then, the gradient flow of the functional $\mathrm{F}_\mu(\rho)$ is given by:

$$
\frac{\partial \rho}{\partial t} = -\nabla \mathrm{F}_\mu(\rho_t),
\tag{2}
$$

where the subscript $t$ refers to the time (iteration) index throughout this work. Equivalently, equation (1) follows a partial differential equation as:

$$
\partial_t \rho_t(\mathbf{v}) - \nabla_{\mathbf{v}} \cdot (\rho_t(\mathbf{v})\nabla_{\mathbf{v}} \log(\rho_t(\mathbf{v})/\mu(\mathbf{v}))) - \gamma\Delta_{\mathbf{vv}}\rho_t(\mathbf{v}) = 0,
\tag{3}
$$

where $\nabla_{\mathbf{v}}$ and $\Delta_{\mathbf{vv}}$ denote the divergence and the Laplace operators respectively (Ansari et al., 2021). For the equation (3), we have the equivalent stochastic differential equation defined as:

$$
d\mathbf{v}_t = -\nabla_{\mathbf{v}} \log(\rho_t(\mathbf{v})/\mu(\mathbf{v}))dt + \sqrt{2\gamma}d\mathbf{w}_t,
\tag{4}
$$

where $\mathbf{w}_t$ follows the standard Wiener process (Risken, 1996).

We can simulate a sample $\mathbf{v}_0 \sim \rho_0 = \rho$ using equation (4) to obtain a sample close to $\mu$. In practice, this simulation is approximated using the Euler-Maruyama method:

$$
\mathbf{v}_{t+1} = \mathbf{v}_t - \eta\nabla_{\mathbf{v}} \log(\rho_t(\mathbf{v})/\mu(\mathbf{v})) + \sqrt{2\gamma\eta}\xi_t,
\tag{5}
$$

where $\eta > 0$ is the step size, $\xi_t \sim \mathcal{N}(\mathbf{0}, \mathbf{I})$, and $t \in [0, T]$ for the predefined number of iterations $T$. Furthermore, we can address intractability of $\mu(\mathbf{v})$ in equation (5) by approximating the density ratio, $\rho_t(\mathbf{v})/\mu(\mathbf{v})$ by $\rho_0(\mathbf{v})/\mu(\mathbf{v})$, as $\rho_t(\mathbf{v})/\mu(\mathbf{v}) \approx \rho_0(\mathbf{v})/\mu(\mathbf{v})$ for small $t$ and $\eta \to 0$. Given a classifier $D_\phi(\mathbf{v})$ (e.g., mask decoder parameterized by $\phi$) that represents the conditional probability of $\mathbf{v}$ being a sample from $\mu$, the following expression provides an approximation for the density ratio (Sugiyama et al., 2012):

$$
\frac{\rho_0(\mathbf{v})}{\mu(\mathbf{v})} = \frac{1 - D_\phi(\mathbf{v})}{D_\phi(\mathbf{v})} = \exp\left(-d_\phi(\mathbf{v})\right).
\tag{6}
$$

Substituting equation (6) into equation (5) yields the following results:

$$
\begin{aligned}
\mathbf{v}_{t+1} &= \mathbf{v}_t - \eta\nabla_{\mathbf{v}} \log\left(\frac{1 - D_\phi(\mathbf{v})}{D_\phi(\mathbf{v})}\right) + \sqrt{2\gamma\eta}\xi_t \\
&= \mathbf{v}_t + \eta\nabla_{\mathbf{v}}d_\phi(\mathbf{v}) + \sqrt{2\gamma\eta}\xi_t,
\end{aligned}
\tag{7}
$$

where $d_\phi(\mathbf{v}) = -log((1 - D_\phi)/D_\phi)$ is the logit output of the classifier $D_\phi$.

In this work, $\rho$ is the distribution of query image embeddings generated by SAM's encoder, $\mu$ represents the distribution of the optimal embedding with respective to the mask decoder, and $D_\phi$ is SAM's mask decoder (i.e., pixel-wise classifier). The $D_\phi$ can take prompts such as bounding boxes and prompt points as inputs in addition to $\mathbf{v}$, we omitted them in equation (6) and equation (7) for clear explanation.

### 3.2 Segmentation refinement via mask decoder gradient flow

Here, we elaborate how gradient flow can be adopted for refining predicted masks. We describe the general framework of few-shot segmentation using promptable segmentation models, and then discuss our method for improving mask quality using gradient flow. As illustrated in Fig. 2, the first

Figure 2: **Overview of ReGRAF**. *Encoder* box denotes image encoder, *sim* box denotes simliarity mesuring module, and *Mask Decoder* box denotes SAM's mask decoder. SAM's prompt encoder is omitted here to show clear flow of our method. ReGRAF is a training-free approach that utilizes a query image, a support image, and its associated mask (blue region on the support image) to generate segmentation masks. First, the *encoder* extracts query embeddings $\mathbf{z}_0^q$ and support embeddings $\mathbf{z}_0^s$. Prompts are obtained based on the similarity between the embeddings $\mathbf{z}_0^q$ and $\mathbf{z}_0^s$. Thereafter, the *mask decoder* of a promptable segmentation model computes logits given the prompts and query image embeddings. The logits $d_\phi$ and the query embeddings are then passed to the *gradient flow module* to update the query embeddings $z_t^q$ to $z_{t+1}^q$. Finally, the refined segmentation mask is obtained using the binary mask decoder $D_\phi^{bin}$ at the last iteration index.

step of the framework is to extract image embeddings of a support image $\mathbf{I}^s$ and a query image $\mathbf{I}^q$ by an image encoder $E_\theta$ as

$$\mathbf{z}_0^s = E_\theta(\mathbf{I}^s), \quad \mathbf{z}_0^q = E_\theta(\mathbf{I}^q), \tag{8}$$

where $\mathbf{z}_0^s$ and $\mathbf{z}_0^q$ are initial image embeddings of $\mathbf{I}^s$ and $\mathbf{I}^q$ respectively. Across an image, equation (8) yields two sets of embeddings, $\{\mathbf{z}_{0,i}^s\}_{i=1}^N$ and $\{\mathbf{z}_{0,i}^q\}_{i=1}^N$ where $N$ is the number of pixels in the embedding space. Then, we compute the initial similarity matrix $S_0$ between the query embeddings and the support embeddings. The $i^{th}$ row and $j^{th}$ column of the matrix, $S_0[i,j]$ is computed as:

$$S_0[i,j] = sim(\mathbf{z}_{0,i}^s, \mathbf{z}_{0,j}^q), \tag{9}$$

where *sim* denotes a similarity function between two vectors, representing the similarity measurement module of the few-shot segmentation model. Finally, we sample prompts $P_0$ representing the position of the target object, and obtain an initially predicted mask $\hat{\mathbf{m}}_0$ using a binary mask decoder $D_\phi^{bin}$:

$$P_0 = prompt\_sampler(S_0), \tag{10}$$

$$\hat{\mathbf{m}}_0 = D_\phi^{bin}(\mathbf{z}_0^q; P_0), \tag{11}$$

where $P_0$ is a set of prompts obtained from the similarity map $S_0$, and examples of *prompt_sampler* include top-k sampling (Zhang et al., 2024) and robust sampler (Liu et al., 2024). During this process, some prompts in the set $P_0$ are false positives or semantically confusing. Several suboptimal methods have been proposed to address this issue, such as cascaded refinement, mask filtering, and robust sampling (Zhang et al., 2024; Liu et al., 2024). While these approaches yield reasonably refined masks, they still leave room for improvement. (see Fig. 1)

Thus, we propose our novel process of aligning prompts with the query image semantics using the gradient flow of the mask decoder. In particular, we enhance mask quality by iteratively updating the query embeddings, ensuring that predicted masks are more closely aligned with samples from the true mask distribution. Formally, we have the following gradient flow in the embedding space:

$$\mathbf{z}_{t+1}^q = \mathbf{z}_t^q + \eta \nabla_{\mathbf{z}_t^q} d_\phi(\mathbf{z}_t^q, P_t) + \sqrt{2\gamma\eta}\xi_t, \tag{12}$$

$$S_{t+1}[i,j] = sim(\mathbf{z}_{0,i}^s, \mathbf{z}_{t+1,j}^q), \tag{13}$$

$$P_{t+1} = prompt\_sampler(S_{t+1}), \tag{14}$$

$$\hat{\mathbf{m}}_T = D_\phi^{bin}(\mathbf{z}_0^q; P_T). \tag{15}$$

At the $t$th iteration, we refine the query embedding $\mathbf{z}_t^q$ through the gradient flow derived from the mask decoder as equation (12). Then we compute the updated similarity matrix $S_{t+1}$ following equation (13), and sample prompts $P_{t+1}$ from the $S_{t+1}$ (equation (14)). Ultimately, we obtain refined

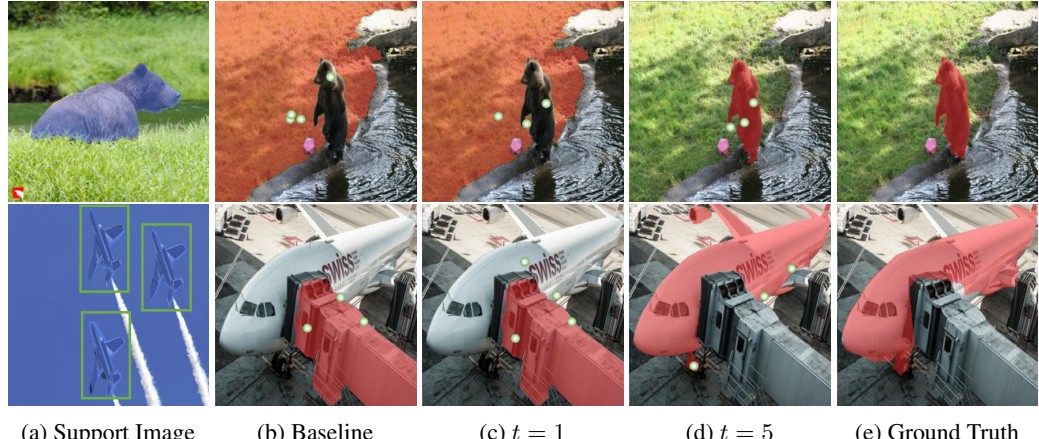

|                     |              |         |         |                  |
| (a) Support Image   | (b) Baseline | (c) $t=1$ | (d) $t=5$ | (e) Ground Truth |

Figure 3: **Refinement process of ReGRAF.** Curated illustrations of ReGRAF's refinement process with PerSAM-F as the baseline. a) Target objects in the support images with blue masks (bear (top) and airplane (bottom)), b) Segmentation results from PerSAM-F with point prompts in green dots. c), d) Refined prompt and segmentation after one and five iterations, e) Ground truth.

mask $\hat{\mathbf{m}}_T$ using the binary mask decoder $D_\phi^{bin}$ (equation (15)). Our overall algorithm is summarized in Alg. 1, and Fig. 3 illustrates the progressive refinement process of our method, where points located at the same position are pruned.

---

**Algorithm 1** Mask refinement via ReGRAF

---

**Require: image encoder ($\theta$), mask decoder ($\phi$), iterations ($T$), step size ($\eta$), noise factor ($\gamma$)**

    Obtain initial image embeddings (equation (8))
    Compute initial similarity and prompts (equation (9)).
    **for** $t = 0$ to $T - 1$ **do**
        Update query embedding (equation (12)).
        Compute similarity and prompts (equation (13, 14))
    **end for**
    Obtain refined mask (equation (15)).
**return** refined mask

---

### 3.3 THEORETICAL ANALYSIS

This section discusses the convergence of ReGRAF. We pose the following key questions:

> *How does the gradient flow approach the optimal probability density function (pdf) $\mu^*$,*
> *and what is the rate at which this convergence occurs?*

This question is crucial for two reasons: First, ReGRAF should be theoretically designed to be a convergent algorithm, which underscores its foundational robustness. Second, considering the approximation of the density ratio used in ReGRAF, even a limited number of iterations should lead to improvements in the quality of segmentation masks. In the following theorem, we demonstrate that ReGRAF addresses the questions above.

**Theorem 1.** *Let $\rho_t$ be a candidate pdf in 2-Wasserstein space $\mathcal{W}_2$ evolving according to the gradient flow of the entropy-regularized KL divergence $F_\mu(\rho)$ with local minimum $\mu^*$. If the initial pdf $\rho_0$ lies in a neighborhood of $\mu^*$, then $\rho_t$ converges to $\mu^*$ exponentially as $t \to \infty$.*

The proof of Theorem 1 is given in Appendix A, where we outline each step to establish the convergence. We assume that the initial probability density function (pdf) $\rho_0$ is *close to* the optimal pdf $\mu^*$. By applying a second-order Taylor expansion around $\mu^*$, we approximate the entropy-regularized KL divergence $F_\mu(\rho)$ which confirms that it reaches its minimum at $\mu^*$. This results in a differential equation demonstrating that the squared distance between $\rho_t$ and $\mu^*$ decreases over time, leading to exponential convergence toward $\mu^*$ as $t$ approaches infinity.

Given that the VFM has been trained on a diverse and extensive dataset, enabling it to generalize across various tasks, it is reasonable to assume that the distribution of the VFM's image embedding $\rho_0$ lies in the neighborhood of the optimal embedding distribution $\mu^*$ for the VFM's mask decoder. Consequently, Theorem 1 guarantees that $\rho_0$ converges to $\mu^*$ at an exponential rate.

In summary, Theorem 1 yields two important implications: 1) it validates the density ratio approximation of ReGRAF, thereby reinforcing the model's theoretical foundation, and 2) it underscores ReGRAF's robustness in refining segmentation masks as a convergent algorithm.

## 4 EXPERIMENTS

### 4.1 EXPERIMENT SETTING

**Baseline method and backbone models.** We utilize PerSAM-F and Matcher as our baseline methods and demonstrate that ReGRAF enhances the segmentation mask for both methods. PerSAM-F uses SAM as its backbone model, fully leveraging it to capture visual clues of target objects in the support images. In contrast, Matcher employs DINOv2 (Oquab et al., 2023) with the ViT-L/14 architecture (Dosovitskiy et al., 2021) as an image encoder to extract embeddings from both query and support images, and uses SAM's decoder as the segmenter. To illustrate that ReGRAF can be seamlessly integrated with various SAM variants, we incorporate HQ-SAM (Ke et al., 2024) into both baselines, resulting in HQ-PerSAM-F and HQ-Matcher. HQ-PerSAM-F substitutes HQ-SAM for SAM, while HQ-Matcher replaces the SAM's mask decoder and prompt encoder with those from HQ-SAM, leaving the other modules unchanged.

**Hyperparameter setting of PerSAM-F/HQ-PerSAM-F.** The hyperparameters for fine-tuning PerSAM-F and HQ-PerSAM-F adhere to the experimental settings outlined in Zhang et al. (2024). We used the SAM with ViT-H as the segmenter for both PerSAM-F and HQ-PerSAM-F, utilizing the AdamW optimizer (Loshchilov, 2017) with a weight decay of 0.01, betas of (0.9, 0.999), and epsilon set to $1e^{-8}$. The learning rate was fixed at $1e^{-3}$, and the model was fine-tuned for 1000 epochs on each query image.

**Hyperparameter setting of Matcher/HQ-Matcher.** We slightly modified the prompt filtering options for Matcher and HQ-Matcher while keeping the other hyperparameters unchanged. This adjustment was made to enhance the robustness of the selection process for accepted masks among the proposal masks generated by Matcher, ultimately reducing the number of accepted masks. The reduction aims to decrease the GPU resources required for computing gradient flow and to minimize optimization errors caused by false positive candidate masks.

**Hyperparameter setting of ReGRAF.** We describe the hyperparameter settings used for ReGRAF. Across all the datasets, we set the number of iterations $T = 5$, as equation (6) implies that the gradient estimation in equation (12) becomes inaccurate with a large number of iterations. The coefficient of the entropy regularization $\gamma$ was set as 0.1 without any tuning process, while the step size $\eta$ was determined from 10 randomly sampled images from left-aside validation sets of COCO-$20^i$ and PACO-part. Specifically, $\eta$ of semantic segmentation was 0.001, and that of part segmentation was 0.0001. Since the tuning for semantic/part segmentation has already been completed during validation, these values were directly applied to other datasets or methods. Additionally, the number of points selected based on similarity (in the case of Matcher, the final number of clustering centers) was set as follows: for semantic segmentation, Matcher used 8 points, HQ-Matcher used 7 points, and PerSAM-F and HQ-PerSAM-F extracted 5 points as prompts to perform the task. For part segmentation, the number of points was reduced to 5 for the Matcher/HQ-Matcher and to 3 for the PerSAM-F/HQ-PerSAM-F, to improve the localization of smaller objects. We used *sim* as proposed by Matcher for Matcher, and as proposed by PerSAM-F for PerSAM-F. Finally, the gradients computed during ReGRAF were clipped to ensure stability throughout the overall process.

**Datasets and evaluation.** We evaluated ReGRAF through two experiments: 1) semantic and 2) part segmentations. The former assesses a broad understanding of objects, while the latter evaluates a fine-grained understanding. Semantic segmentation performance of ReGRAF was assessed on three datasets: FSS-1000 (Li et al., 2020), LVIS-$92^i$ (Liu et al., 2024), and COCO-$20^i$ (Nguyen & Todorovic, 2019). For the part segmentation, we used two datasets: PASCAL-Part (Everingham et al., 2010; Chen et al., 2014; Li et al., 2020) and PACO-Part (Liu et al., 2024; Ramanathan et al., 2023). For all datasets, we adhered to the data preprocessing and evaluation protocols introduced in Liu et al. (2024), and additionally reported the few-shot (5-shot) segmentation performance of

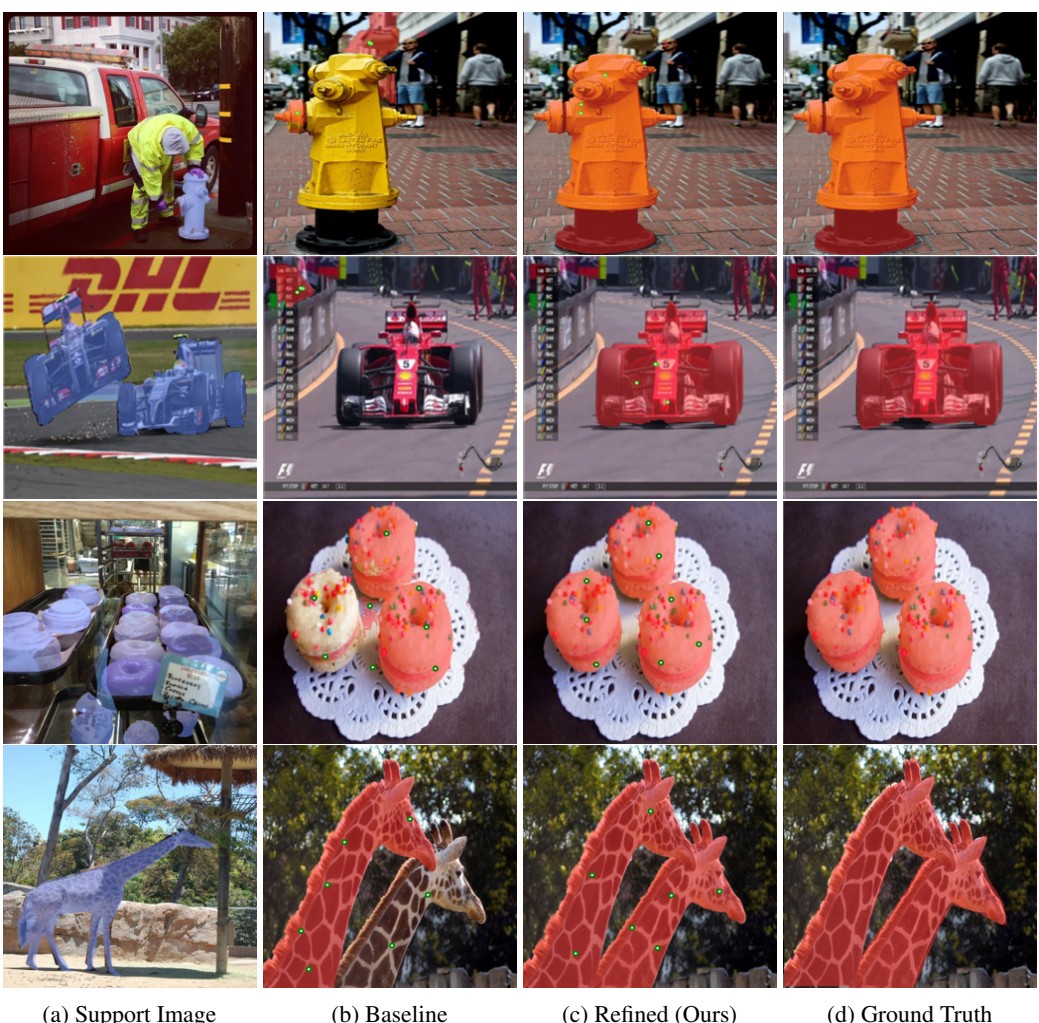

|  (a) Support Image | (b) Baseline | (c) Refined (Ours) | (d) Ground Truth |

Figure 4: **Qualitative result of semantic segmentation**. Refinement process of ReGRAF with various baselines, presented in the following order (top to bottom) : PerSAM-F, HQ-PerSAM-F, Matcher, and HQ-Matcher. Target objects in support images are highlighted with blue masks, and point prompts are denoted by green dots.

Matcher/HQ-Matcher and ReGRAF paired with Matcher/HQ-Matcher. To verify the effectiveness of our method, we measured mean Intersection of Union (mIoU) and also listed the mIoU gain of ReGRAF from baselines across iterations ($T = 5$). The best average mIoU of each dataset within the total iterations of both semantic and part segmentation are reported in Tab. 1 and Tab. 2, while the average mIoU progression and the *oracle* results are presented in Appendix D. Furthermore, we conducted the sensitivity analysis on our hyperparameters ($\eta$ and $T$) in Appendix F.

### 4.2 EXPERIMENT RESULT

We present comprehensive experimental results for semantic and part segmentations, along with a discussion of the additional computational costs associated with our method. For each type of task, one-shot segmentation was performed for PerSAM-F and HQ-PerSAM-F, while both one-shot and few-shot (5-shot) segmentation were performed for Matcher and HQ-Matcher. Experiments on the former were conducted using an NVIDIA GeForce RTX 3090, while the latter were tested on an NVIDIA RTX A6000. Due to space limitations, The qualitative results of semantic and part segmentation for each baseline method, as well as the qualitative results for 5-shot segmentation, are included in Appendix B. Additionally, some failure cases of ReGRAF is provided in Appendix C.

#### 4.2.1 SEMANTIC SEGMENTATION.

To evaluate the model's comprehensive understanding of a scene, we conducted a comparative analysis of ReGRAF with baseline methods in semantic segmentation. We evaluated ReGRAF on three

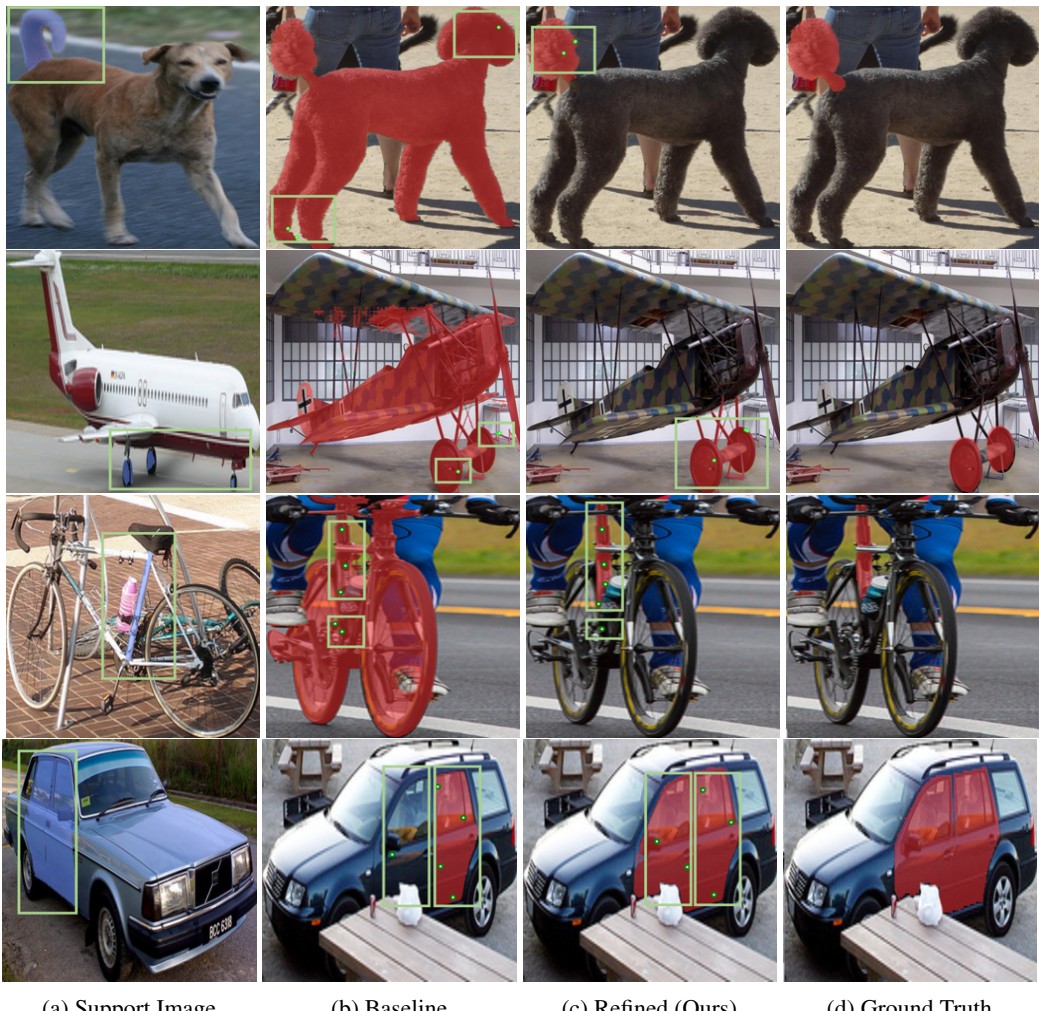

(a) Support Image  (b) Baseline  (c) Refined (Ours)  (d) Ground Truth

Figure 5: **Qualitative result of part segmentation**. Refinement process of ReGRAF with various baselines, presented in the following order (top to bottom) : PerSAM-F, HQ-PerSAM-F, Matcher, and HQ-Matcher. Target objects in the support images are highlighted with blue masks and enclosed in green boxes, while point prompts are denoted by green dots and also emphasized within green boxes.

datasets: COCO-20$^i$, FSS-1000, and LVIS-92$^i$, all of which were preprocessed to ensure consistent, mask-annotated segmentation tasks. We followed the established evaluation schemes on COCO-20$^i$ and FSS-1000, and tested all baseline methods directly on their respective test sets. Additionally, LVIS-92$^i$ was curated as a challenging benchmark to test cross-dataset generalization with balanced class folds and randomly sampled evaluation episodes. Tab. 1 quantitatively demonstrates consistent mIoU improvements across a variety of experimental settings. These results demonstrate ReGRAF's effectiveness in refining segmentation mask quality in diverse one/few-shot scenarios. The enhance-

Table 1: Semantic segmentation. (mIoU)

| Baseline (#-shot) | FSS-1000 | | COCO-20$^i$ | | LVIS-92$^i$ | |
|---|---|---|---|---|---|---|
| | Base | + ReGRAF | Base | + ReGRAF | Base | + ReGRAF |
| **PerSAM-F** | 50.90 | 54.47 +3.57 | 16.12 | 16.59 +0.47 | 7.30 | 7.78 +0.49 |
| **HQ-PerSAM-F** | 69.70 | 72.17 +2.47 | 24.84 | 25.17 +0.34 | 10.88 | 10.99 +0.11 |
| **Matcher** | 92.07 | 92.09 +0.02 | 69.80 | 70.45 +0.66 | 62.13 | 62.52 +0.40 |
| **HQ-Matcher** | 92.50 | 92.80 +0.30 | 70.06 | 70.55 +0.49 | 60.04 | 60.37 +0.34 |
| **Matcher (5)** | 93.08 | 93.26 +0.18 | 67.61 | 68.48 +0.88 | 57.12 | 58.08 +0.97 |
| **HQ-Matcher (5)** | 93.25 | 93.29 +0.05 | 67.77 | 68.20 +0.43 | 57.44 | 58.14 +0.71 |

Table 2: Part segmentation. (mIoU)

| Baseline (#-shot) | PACO-part | | Pascal-part | |
|---|---|---|---|---|
| | Base | + ReGRAF | Base | + ReGRAF |
| **PerSAM-F** | 19.39 | 19.68 +0.29 | 24.44 | 24.57 +0.13 |
| **HQ-PerSAM-F** | 20.84 | 20.91 +0.08 | 26.96 | 27.05 +0.09 |
| **Matcher** | 50.25 | 50.33 +0.08 | 54.61 | 54.91 +0.31 |
| **HQ-Matcher** | 51.13 | 51.32 +0.19 | 56.23 | 56.46 +0.23 |
| **Matcher (5)** | 48.70 | 48.84 +0.14 | 54.50 | 54.66 +0.17 |
| **HQ-Matcher (5)** | 49.39 | 49.66 +0.27 | 56.30 | 56.40 +0.10 |

ments in semantic segmentation quality reflect ReGRAF's comprehensive understanding of the scene. As illustrated in Fig. 4, baseline methods occasionally struggle to segment the target object in the

query image, primarily due to false positives resulting from a limited understanding of the target object's semantics. In such scenarios, baseline approaches often fail to accurately delineate the target object's outline (1st row in Fig. 4), localize it effectively due to dominant features such as color (2nd row in Fig. 4), or generate diverse prompts that fully capture entire target objects in a scene (3rd and 4th row in Fig. 4). In contrast, ReGRAF effectively resolves these challenges within just five refinement iterations.

### 4.2.2 PART SEGMENTATION.

Part segmentation assesses how effectively a model comprehends fine-grained semantics and accurately performs segmentation tasks. We measured the gains in mIoU by ReGRAF from each baseline methods on two datasets, PASCAL-part and PACO-part. Both datasets are constructed in the work of Liu et al. (2024) to create a robust framework for evaluating part segmentation models, utilizing cropped objects with bounding boxes to focus on segmentation quality, facilitating more precise one-shot segmentation tasks.

As shown in Tab. 2, ReGRAF makes consistent refinement of segmentation masks across different baselines. While the gains in mIoU may appear modest, note that part segmentation presents unique challenges due to the granularity and complexity of the object details. Despite limited numerical improvements in some cases, ReGRAF excels in capturing the delicate structure of the fine grained target part. While the outputs from various baseline methods often struggle to align precisely with the delicate structures of the target parts (e.g. placing some prompts on the object containing the part, rather than on the part itself (1st and 3rd row in Fig. 5), mislocalizing prompts (2nd row in Fig. 5), or failing to capture the entire part (the last row in Fig. 5)), ReGRAF effectively aligns prompts with the fine-grained parts of the target objects, as shown in the second column of Fig. 5. This enhanced alignment results from ReGRAF's capability to refine the query image embeddings in relation to a VFM's mask decoder, which leads to more accurate prompt localization.

### 4.2.3 DISCUSSION ON COMPUTATION OF REGRAF.

ReGRAF requires at least 20GB and 8GB GPU memory when used with Matcher and PerSAM-F respectively. We argue that the advantages of our approach outweigh the associated memory consumption because ReGRAF enhances the quality of segmentation masks without requiring additional learnable parameters and new training datasets for fine-tuning or modifications to the model architecture as in previous approaches. This efficiency not only reduces the complexity of the implementation but also facilitates broader applicability across various segmentation frameworks.

Moreover, the additional time required for gradient flow computation per image is minimal compared to the total iteration time. For instance, the gradient update takes only 0.02 seconds per iteration, while the running time for PerSAM-F with ReGRAF is 8.02 seconds (Tab. 3). Thus, the computational overhead introduced by our method is negligible, further reinforcing its practicality and effectiveness in enhancing segmentation mask quality.

Table 3: Per-image running time of baselines and ReGRAF

| Method | Running time of baselines (sec) | ReGRAF (sec/iteration) |
|---|---|---|
| PerSAM-F+ReGRAF | 8.02 | **0.02** |
| Matcher+ReGRAF | 2.16 | **0.08** |

## 5 CONCLUSION

In this paper, we introduce a novel training-free refinement method that enhances the quality of segmentation masks using a promptable segmentation model. Our method is widely applicable to auto-prompting frameworks, and we comprehensively evaluated the effectiveness of our approach through extensive quantitative and qualitative assessments, demonstrating its superiority. ReGRAF effectively refines visual clues for SAM's mask decoder, allowing it to understand the semantics of target objects within a scene. Furthermore, theoretical analysis on the convergence of ReGRAF and the experiment results emphasize the robustness and adaptability of our method across both semantic and part segmentation tasks. Our methodology stands out from existing approaches by eliminating the need for learnable parameters, modifications to model architecture, and reliance on additional training datasets. This is highly efficient, as it requires significantly less time for refinement. These advantages highlight the practicality and effectiveness of our method in enhancing segmentation mask quality without the complexities.

## 6 REPRODUCIBILITY

Hyperparameters of each baselines and ReGRAF are described in Sec 4.1, and we will make the code public that produce the same result in this paper. In addition, to ensure the reproducibility of our work, we provided a detailed figure of ReGRAF in Fig. 2 along with the pseudo code of our general framework in Alg. 1. The key script of the code is included in the supplementary materials.

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

# A    PROOF OF THEOREM 1

**Theorem 1.** *Let $\rho_t$ be a candidate pdf in 2-Wasserstein space $\mathcal{W}_2$ evolving according to the gradient flow of the entropy-regularized KL divergence $F_\mu(\rho)$ with local minimum $\mu^*$. If the initial pdf $\rho_0$ lies in a neighborhood of $\mu^*$, then $\rho_t$ converges to $\mu^*$ exponentially as $t \to \infty$.*

*Proof.* To prove Theorem 1, We begin by the assumption that $\rho_0$ is a neighborhood of $\mu^*$, and approximate $F_\mu(\rho)$ near $\mu^*$ by the second Taylor expansion as:

$$
\begin{aligned}
F_\mu(\rho) &= F_\mu(\mu^*) + \nabla F_\mu(\mu^*)(\rho - \mu^*) + \frac{1}{2} \left\langle F_\mu^{''}(\mu^*)(\rho - \mu^*), \rho - \mu^* \right\rangle + O\left( \|\rho_t - \mu^*\|^3 \right) \\
&= F_\mu(\mu^*) + \frac{1}{2} \left\langle F_\mu^{''}(\mu^*)(\rho - \mu^*), \rho - \mu^* \right\rangle,
\end{aligned}
\tag{16}
$$

where $\langle \, , \, \rangle$ is an inner product operator, $O$ is the big-O notation, and the second line of equation (16) holds because $\mu^*$ is the minimizer of $F_\mu$ (i.e. $\nabla F_\mu(\mu^*) = 0$). Furthermore, since $\mu^*$ is the minimizer of the functional $F_\mu$, the Hessian $F_\mu^{''}$ is positive definite.

Differentiating $F_\mu(\rho_t)$ with respective to $\rho_t$ in equation (16), we obtain the following approximation for $\nabla F_\mu(\rho_t)$:

$$
\nabla F_\mu(\rho_t) \approx F_\mu^{''}(\mu^*)(\rho_t - \mu^*).
\tag{17}
$$

The above approximation reduces equation (2) to:

$$
\frac{\partial \rho_t}{\partial t} = -F_\mu^{''}(\mu^*)(\rho_t - \mu^*).
\tag{18}
$$

To show that equation (18) describes a linear system with exponential convergence, consider the following differential equation:

$$
\begin{aligned}
\frac{d}{dt} \|\rho_\tau - \mu^*\|^2 &= 2 \left\langle \rho_\tau - \mu^*, \frac{\partial \rho_\tau}{\partial \tau} \right\rangle \\
&= 2 \left\langle \rho_\tau - \mu^*, -F_\mu^{''}(\mu^*)(\rho_\tau - \mu^*) \right\rangle,
\end{aligned}
\tag{19}
$$

where $\tau \geq 0$. The second line of the above equation holds by equation (17). Using the positive definiteness of $F_\mu^{''}(\mu^*)$, the following equation holds for the minimal eigenvalue $\lambda_{min}$ of $F_\mu^{''}(\mu^*)$,

$$
\left\langle \rho_\tau - \mu^*, F_\mu^{''}(\mu^*)(\rho_\tau - \mu^*) \right\rangle > \lambda_{min} \|\rho_\tau - \mu^*\|^2.
\tag{20}
$$

Considering equation (19) and equation (20), we obtain:

$$
\frac{d}{dt} \|\rho_\tau - \mu^*\|^2 < -2\lambda_{min} \|\rho_\tau - \mu^*\|^2.
\tag{21}
$$

By dividing both sides of equation (20) by $\|\rho_\tau - \mu^*\|^2$ and integrating from $\tau = 0$ to $\tau = t$, we have:

$$
\log \|\rho_t - \mu^*\|^2 < -2\lambda_{min} t + \log \|\rho_0 - \mu^*\|^2.
\tag{22}
$$

Exponentiating both sides yield:

$$
\|\rho_t - \mu^*\|^2 < e^{-2\lambda_{min} t} \|\rho_0 - \mu^*\|^2,
\tag{23}
$$

which implies that $\rho_t$ converges to $\mu^*$ exponentially as $t \to \infty$ when $\rho_0$ is a neighborhood of $\mu^*$. $\quad\square$

# B  ADDITIONAL QUALITATIVE RESULTS

## B.1  ONE SHOT SEMANTIC SEGMENTATION

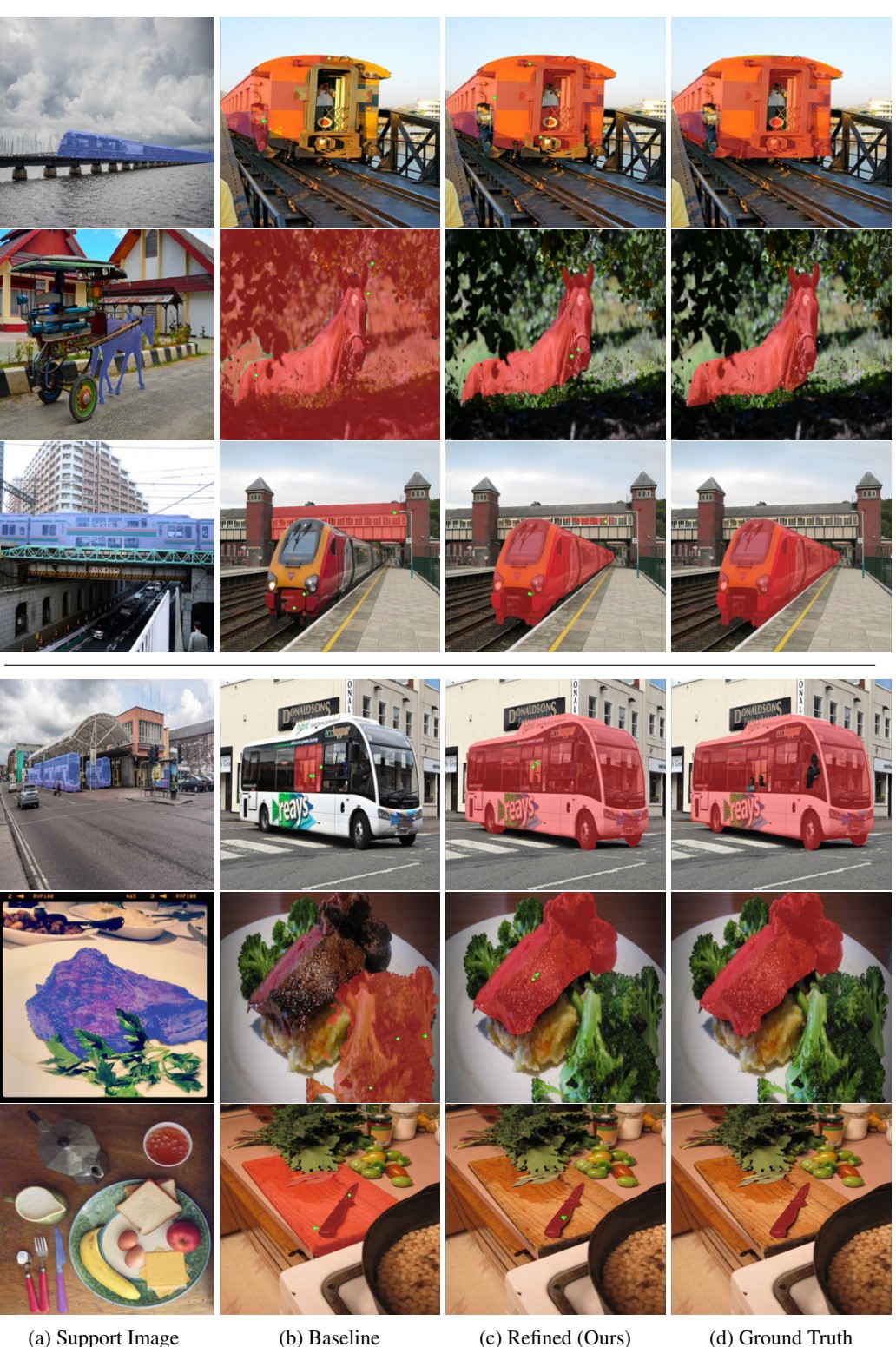

(a) Support Image     (b) Baseline     (c) Refined (Ours)     (d) Ground Truth

Figure 6: Qualitative result of ReGRAF with PerSAM-F/ HQ-PerSAM-F in semantic segmentation. The upper section: ReGRAF with PerSAM-F, and lower section : ReGRAF with HQ-PerSAM-F.

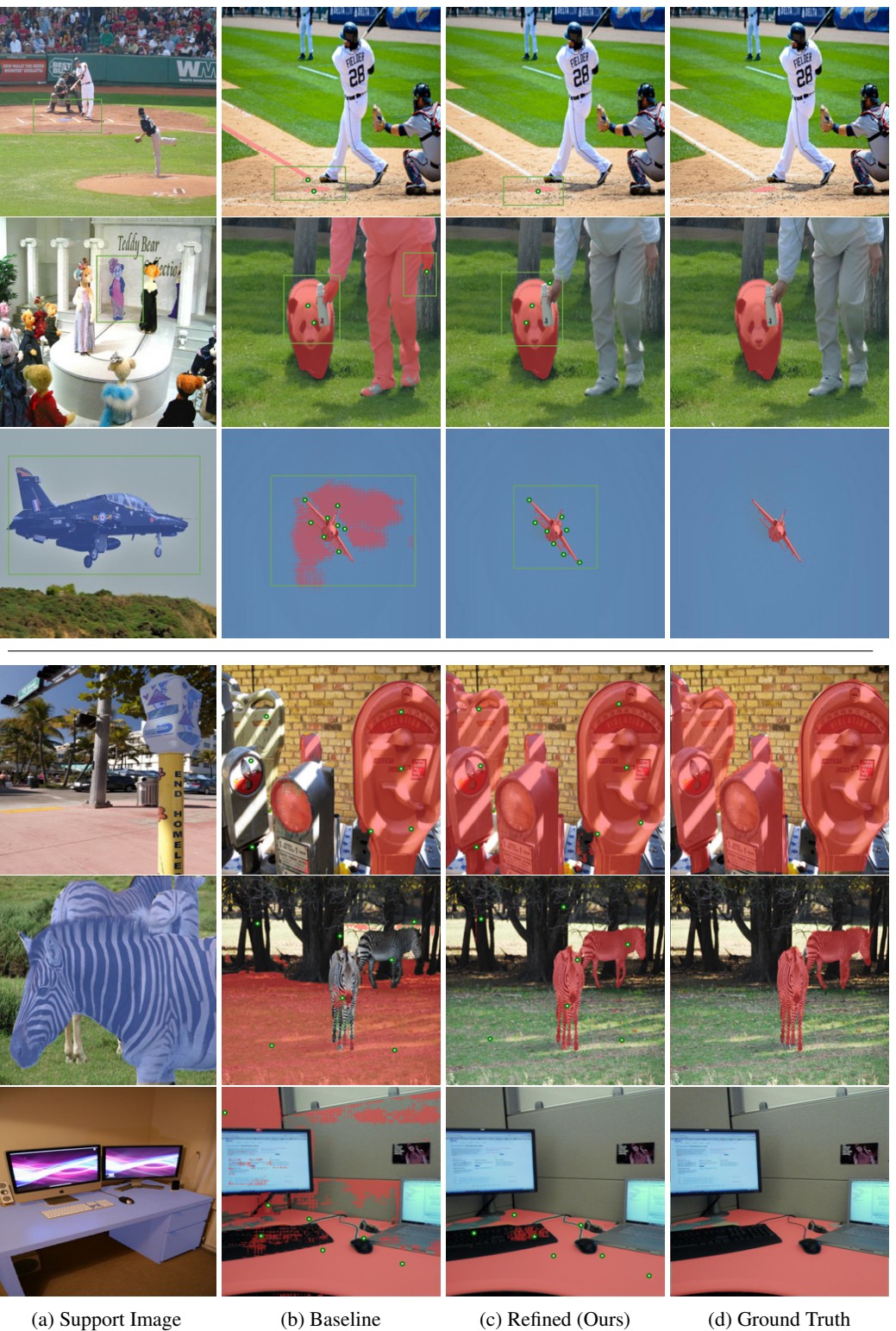

(a) Support Image      (b) Baseline      (c) Refined (Ours)      (d) Ground Truth

Figure 7: Qualitative result of ReGRAF with Matcher/ HQ-Matcher in semantic segmentation. The upper section shows ReGRAF with Matcher, while the lower section displays ReGRAF with HQ-Matcher.

## B.2 ONE SHOT PART SEGMENTATION

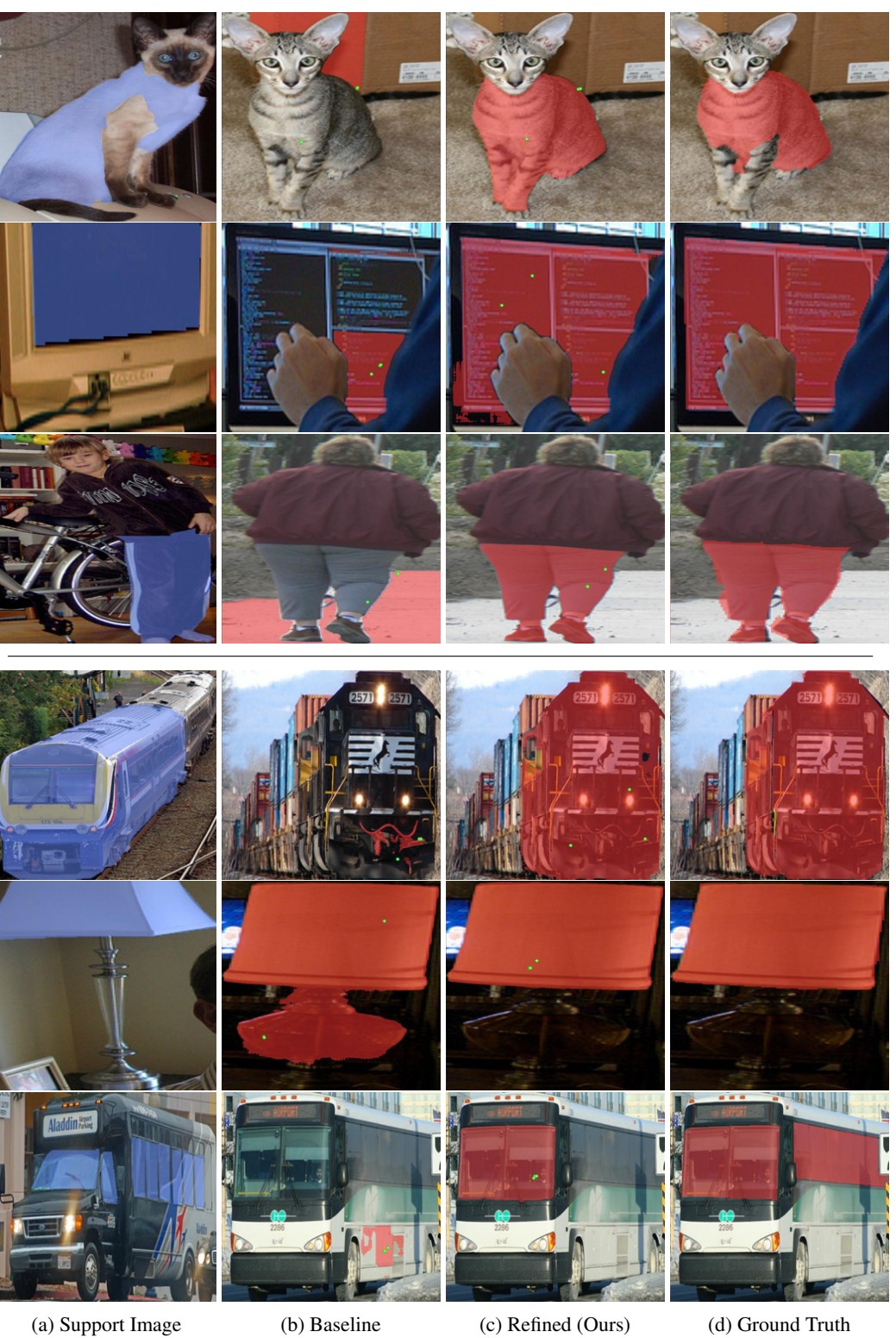

(a) Support Image     (b) Baseline     (c) Refined (Ours)     (d) Ground Truth

Figure 8: Qualitative result of ReGRAF with PerSAM-F/ HQ-PerSAM-F in part segmentation. The upper section shows ReGRAF with PerSAM-F, while the lower section displays ReGRAF with HQ-PerSAM-F.

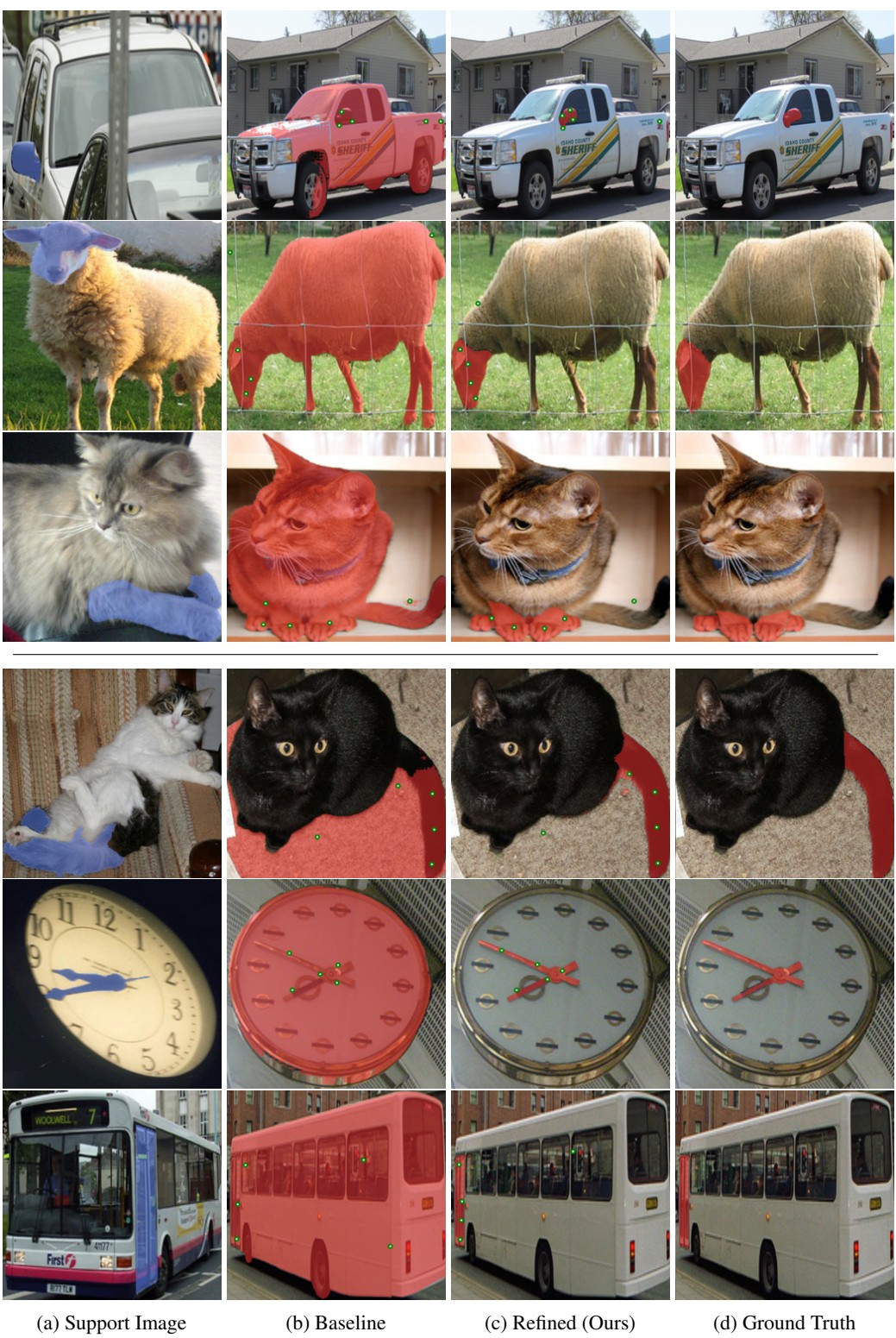

(a) Support Image      (b) Baseline      (c) Refined (Ours)      (d) Ground Truth

Figure 9: Qualitative result of ReGRAF with Matcher/ HQ-Matcher in part segmentation. The upper section shows ReGRAF with Matcher, while the lower section displays ReGRAF with HQ-Matcher.

### B.3 FIVE SHOT SEMANTIC SEGMENTATION (MATCHER)

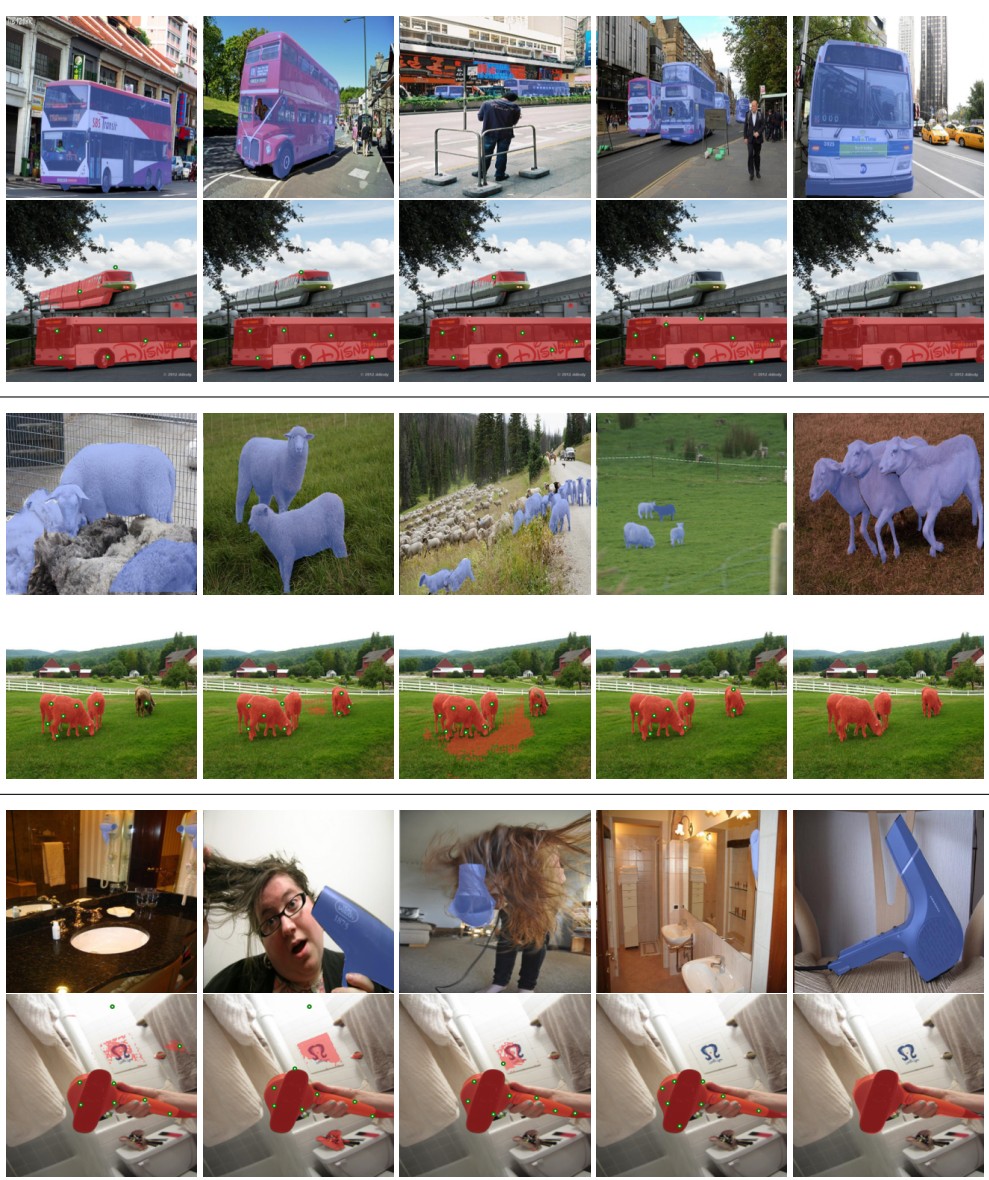

Figure 10: Qualitative result of five shot semantic segmentation (Matcher). The test samples are separated by dividers. For each sample, the upper row shows the support images, while the lower row displays, from left to right, the baseline result, the result of ReGRAF across iterations, and the ground truth.

## B.4 FIVE SHOT SEMANTIC SEGMENTATION (HQ-MATCHER)

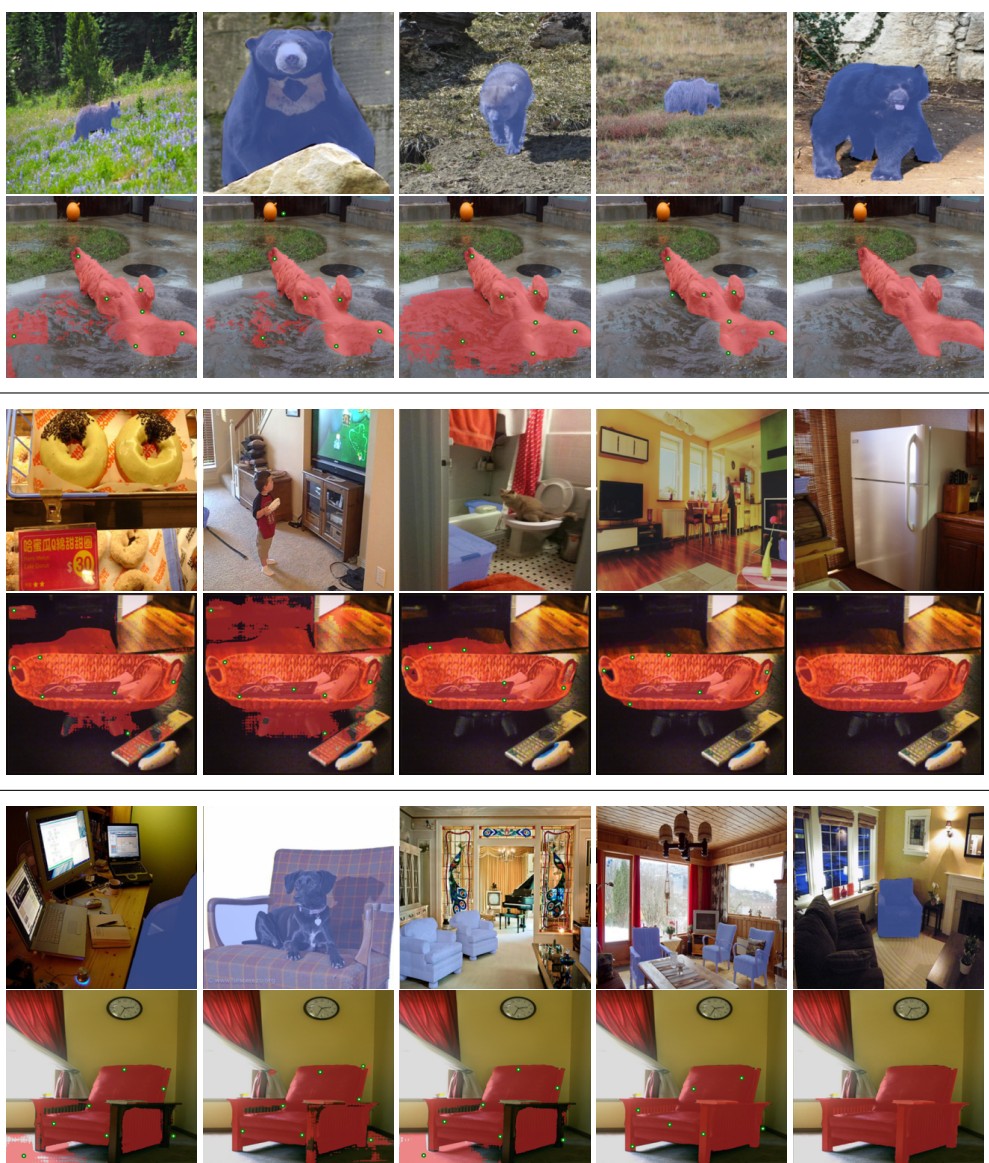

Figure 11: Qualitative result of five shot semantic segmentation (HQ-Matcher). The test samples are separated by dividers. For each sample, the upper row shows the support images, while the lower row displays, from left to right, the baseline result, the result of ReGRAF across iterations, and the ground truth.

B.5 FIVE SHOT PART SEGMENTATION (MATCHER)

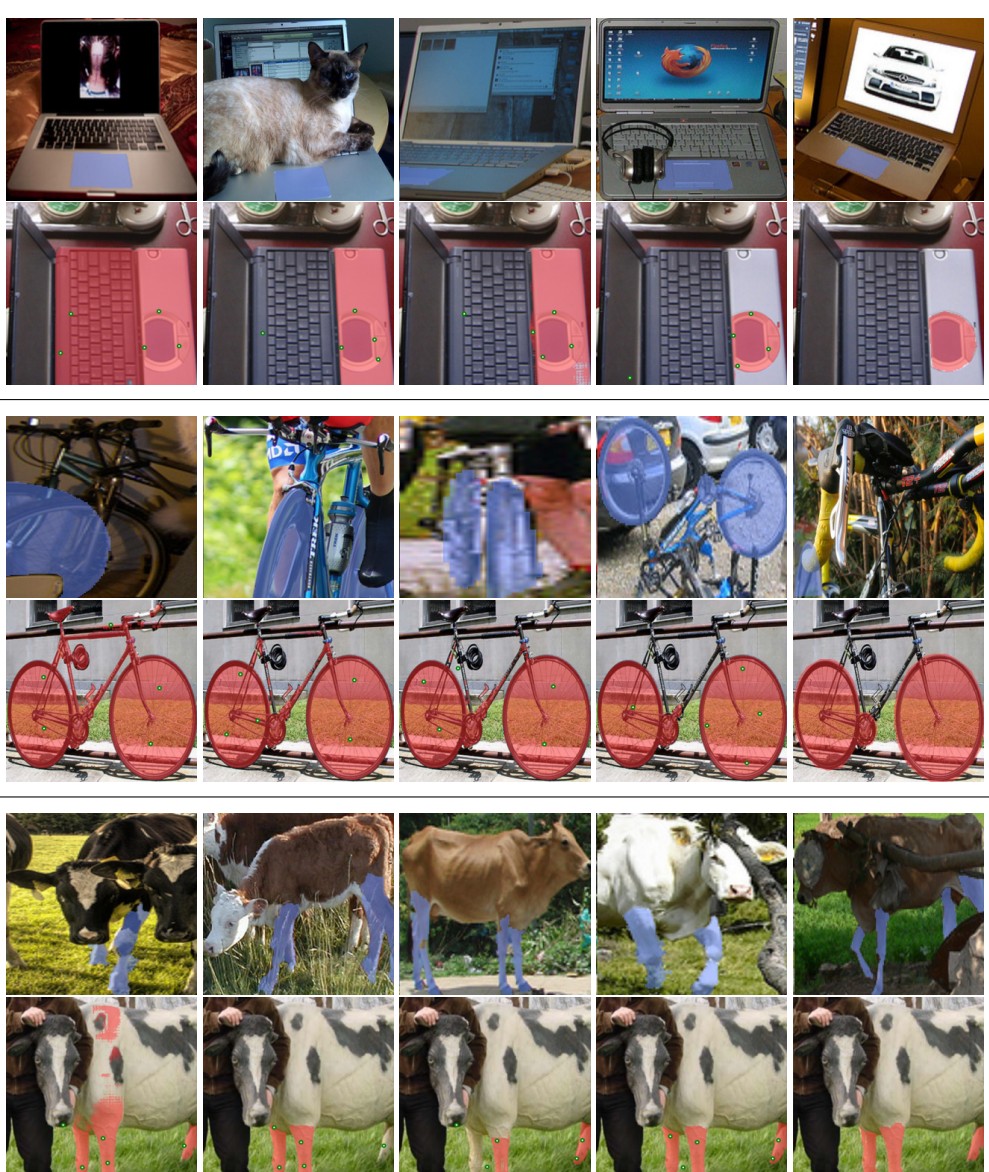

Figure 12: Qualitative result of five shot part segmentation (Matcher). The test samples are separated by dividers. For each sample, the upper row shows the support images, while the lower row displays, from left to right, the baseline result, the result of ReGRAF across iterations, and the ground truth.

## B.6 FIVE SHOT PART SEGMENTATION (HQ-MATCHER)

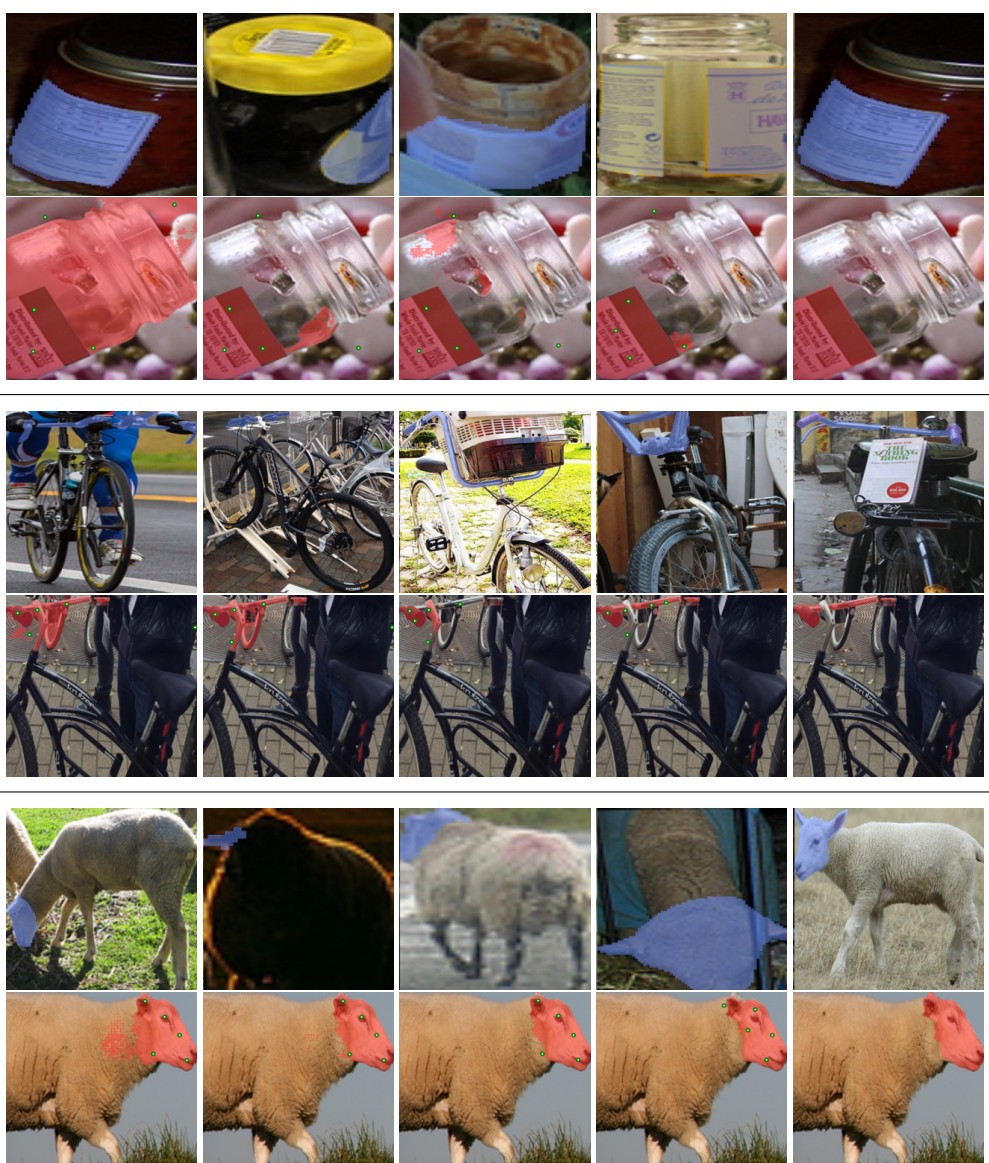

Figure 13: Five shot part segmentation (HQ-Matcher). The test samples are separated by dividers. For each sample, the upper row shows the support images, while the lower row displays, from left to right, the baseline result, the result of ReGRAF across iterations, and the ground truth.

## B.7 COMPARISON OF THE EFFECTS OF REGRAF ACROSS DIFFERENT BASELINES.

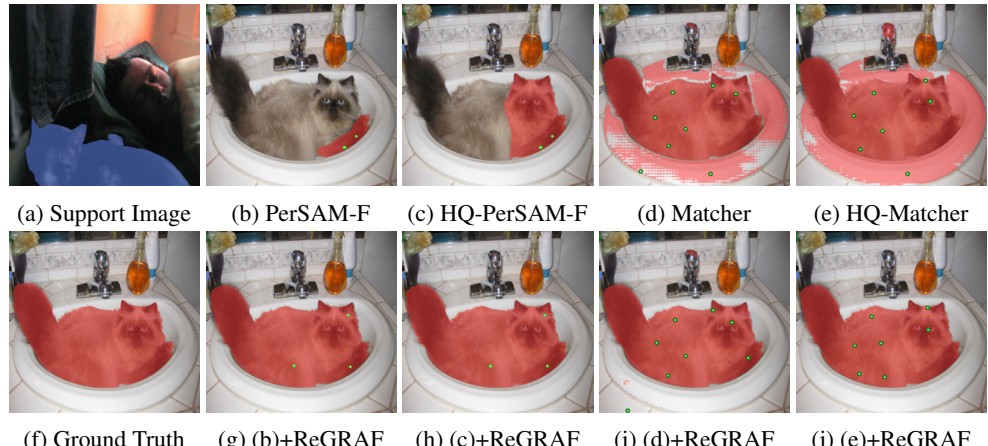

(a) Support Image   (b) PerSAM-F   (c) HQ-PerSAM-F   (d) Matcher   (e) HQ-Matcher

(f) Ground Truth   (g) (b)+ReGRAF   (h) (c)+ReGRAF   (i) (d)+ReGRAF   (j) (e)+ReGRAF

Figure 14: Illustration of prompt refinement of ReGRAF

## C    FAILURE CASES OF ReGRAF

In this section, we demonstrate certain challenging scenarios where prompt refinement is less effective, helping to identify areas for improvement in the method's robustness. When the visual semantics between the reference and target images differ significantly (e.g., the $1st$ and $4th$ rows of Fig. 15), prompt refinement becomes challenging, often resulting in performance degradation. Furthermore, when the visual clues in the reference images are ambiguous—such as difficulty in distinguishing specific parts of a bicycle, or when the reference depicts a general tray while the segmentation target focuses on the tray's edge—ReGRAF encounters challenges in refining the prompts effectively (e.g., the $2nd$, $3rd$, and $5th$ rows of Fig. 15)

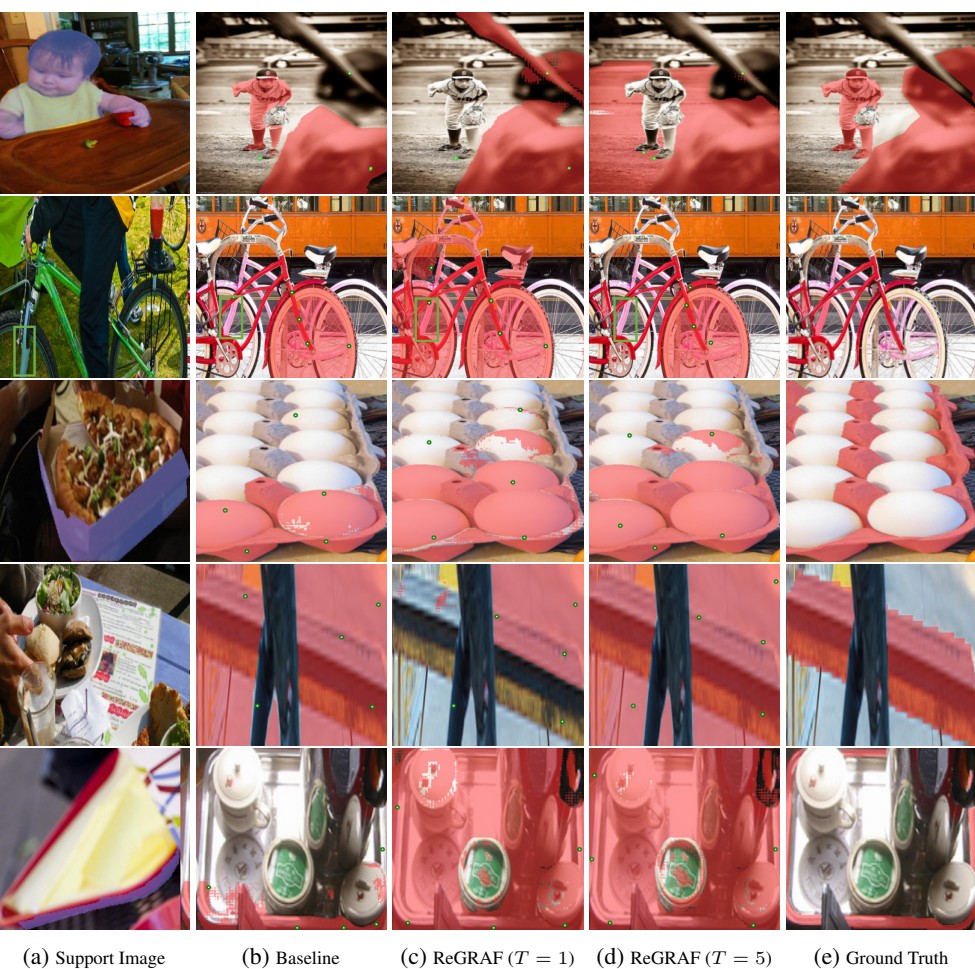

(a) Support Image    (b) Baseline    (c) ReGRAF ($T = 1$)    (d) ReGRAF ($T = 5$)    (e) Ground Truth

Figure 15: Failure cases of ReGRAF. The upper two rows compare ReGRAF with PerSAM-F as the baseline, while the lower three rows compare ReGRAF with Matcher as the baseline. Each comparison illustrates the failure cases of ReGRAF.

# D  AVERAGE MIOU PROGRESSION AND ORACLE RESULTS OF REGRAF

In the following, we present average mIoU segmentation results of each dataset as well as the *oracle* results. The oracle results are obtained by selecting the most accurate segmentation masks across the maximum iterations ($T = 5$) based on a comparison with the ground truth. This approach closely resembles real-world usage of ReGRAF, as it allows selecting the most suitable segmentation result from different iterations.

| Baselines (#-shot) | Datasets | Baseline mIoU | Iterations (ReGRAF Gain) | | | | | Oracle |
|---|---|---|---|---|---|---|---|---|
| | | | 1 | 2 | 3 | 4 | 5 | |
| PerSAM-F | COCO-20i | 16.12 | 0.00 | 0.02 | 0.16 | 0.47 | 0.26 | **7.52** |
| HQ-PerSAM-F | | 24.84 | 0.00 | 0.33 | 0.18 | 0.24 | 0.25 | **5.29** |
| Matcher | | 69.80 | -0.15 | 0.21 | 0.07 | 0.31 | 0.66 | **7.09** |
| HQ-Matcher | | 70.06 | 0.08 | 0.03 | 0.25 | 0.36 | 0.49 | **7.22** |
| Matcher (5) | | 67.61 | 0.17 | -0.20 | -0.05 | 0.50 | 0.88 | **7.98** |
| HQ-Matcher (5) | | 67.78 | -0.22 | -0.08 | -0.15 | -0.04 | 0.42 | **7.95** |
| PerSAM-F | LVIS-92i | 7.30 | 0.00 | 0.18 | 0.23 | 0.44 | 0.48 | **4.26** |
| HQ-PerSAM-F | | 10.88 | 0.00 | 0.05 | 0.11 | 0.05 | 0.06 | **2.93** |
| Matcher | | 62.13 | -0.06 | -0.24 | -0.12 | 0.31 | 0.40 | **6.41** |
| HQ-Matcher | | 60.04 | 0.09 | 0.07 | 0.15 | 0.18 | 0.34 | **6.41** |
| Matcher (5) | | 57.12 | -0.05 | -0.04 | 0.17 | 0.54 | 0.97 | **7.54** |
| HQ-Matcher (5) | | 57.44 | -0.11 | -0.13 | 0.09 | 0.33 | 0.70 | **7.54** |
| PerSAM-F | FSS-1000 | 50.90 | 0.00 | 0.99 | 2.06 | 2.73 | 3.57 | **15.62** |
| HQ-PerSAM-F | | 69.70 | 0.00 | 1.06 | 1.10 | 1.74 | 2.47 | **9.62** |
| Matcher | | 92.07 | -0.19 | -0.10 | 0.02 | -0.05 | -0.05 | **1.42** |
| HQ-Matcher | | 92.50 | 0.15 | 0.17 | 0.30 | 0.16 | 0.13 | **1.60** |
| Matcher (5) | | 93.21 | 0.11 | 0.01 | 0.09 | 0.00 | -0.01 | **1.15** |
| HQ-Matcher (5) | | 93.25 | 0.02 | 0.02 | -0.03 | 0.02 | 0.04 | **1.15** |
| PerSAM-F | PACO-part | 19.39 | 0.00 | 0.13 | 0.29 | 0.18 | 0.22 | **4.67** |
| HQ-PerSAM-F | | 20.84 | 0.00 | -0.13 | -0.05 | -0.03 | 0.07 | **3.66** |
| Matcher | | 64.31 | 0.07 | 0.08 | -0.03 | -0.14 | -0.20 | **6.38** |
| HQ-Matcher | | 51.13 | 0.19 | 0.11 | 0.13 | -0.11 | 0.00 | **6.96** |
| Matcher (5) | | 64.31 | 0.07 | 0.08 | -0.03 | -0.14 | -0.20 | **6.48** |
| HQ-Matcher (5) | | 49.39 | 0.15 | -0.22 | 0.14 | 0.14 | 0.27 | **7.14** |
| PerSAM-F | PASCAL-part | 24.44 | 0.00 | 0.00 | 0.05 | 0.12 | 0.13 | **9.52** |
| HQ-PerSAM-F | | 26.96 | 0.00 | -0.03 | -0.09 | 0.06 | 0.09 | **8.53** |
| Matcher | | 54.62 | 0.05 | 0.00 | 0.30 | 0.04 | -0.04 | **6.49** |
| HQ-Matcher | | 56.23 | -0.06 | -0.07 | -0.12 | 0.07 | 0.23 | **6.49** |
| Matcher (5) | | 54.50 | -0.21 | 0.17 | 0.04 | -0.08 | -0.26 | **6.93** |
| HQ-Matcher (5) | | 57.63 | -0.10 | -0.03 | 0.19 | 0.06 | 0.03 | **6.93** |

Table 4: Average mIoU progression and oracle results of ReGRAF.

Tab. 4 demonstrates that while the optimal iteration may vary across samples, ReGRAF consistently refines segmentation masks effectively, yielding significant performance improvements in practical applications (e.g., approximately a 10% mIoU gain for Matcher and HQ-Matcher).

# E    SEGMENTATION RESULTS ON FINE-GRAINED DATASET

To further validate the performance of ReGRAF on fine-grained objects, we also tested it on the DIS5K dataset (Qin et al., 2022). DIS5K is specifically designed for segmentation tasks where fine-grained objects are difficult to distinguish and require more accurate segmentation masks. We applied the same hyperparameter settings for ReGRAF as those used in part segmentation.

Tab. 5 shows that ReGRAF performs well even on the challenging data. Although the progression of average mIoU gains suggests challenges in selecting the optimal maximum iterations $T$, the oracle results demonstrate that our method effectively enhances segmentation quality, even for this difficult task.

| Baselines | Datasets | Baseline mIoU | Iterations (ReGRAF Gain) | | | | | Oracle |
|---|---|---|---|---|---|---|---|---|
| | | | 1 | 2 | 3 | 4 | 5 | |
| **PerSAM-F** | | 27.47 | 0.00 | 0.24 | 0.33 | 0.02 | -0.24 | **6.33** |
| **HQ-PerSAM-F** | DIS5K | 52.62 | 0.00 | -0.12 | 0.06 | 0.17 | 0.11 | **5.55** |
| **Matcher** | | 46.77 | -0.29 | 0.14 | -0.19 | -0.23 | 0.14 | **10.87** |
| **HQ-Matcher** | | 58.39 | 0.09 | 0.58 | -0.06 | 0.18 | 0.52 | **10.65** |

Table 5: Segmentation results of ReGRAF on DIS5K.

# F    SENSITIVITY ANALYSIS OF THE STEP SIZE AND ITERATIONS

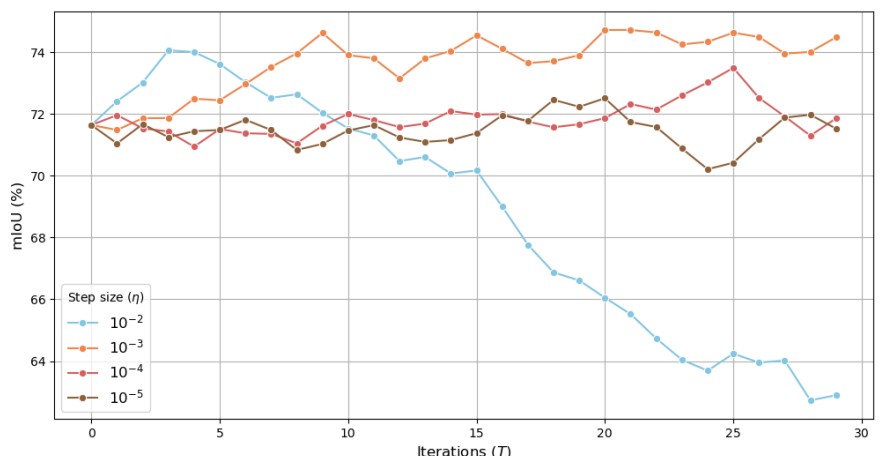

Figure 16: Sensitivity analysis of the step size $\eta$ and iterations $T$.

We randomly sampled 100 examples from the set-aside dataset (derived from the COCO-20i training dataset) to conduct the sensitivity analysis on the step size ($\eta$) and the number of iterations ($T$). For the analysis, we evaluated the mIoU over iterations, testing up to 30 iterations with step sizes of $10^{-2}$, $10^{-3}$, $10^{-4}$, and $10^{-5}$. Fig. 16 shows a moving average with a window size of 2 to smooth fluctuations and better reveal overall trends.

The result highlights the significant impact of the step size ($\eta$) on the segmentation performance and stability. Among the tested values, $\eta = 10^{-3}$ demonstrated the most consistent and superior performance. In contrast, $\eta = 10^{-2}$ showed rapid initial improvement but suffered from performance degradation over time, indicating a lack of long-term stability. Smaller step sizes, such as $\eta = 10^{-4}$ and $\eta = 10^{-5}$, exhibited increased variability and slower convergence.

This underscore the importance of tuning the step size, however ReGRAF can be easily tuned and applicable to various scenarios. As shown in Tab. 3, incorporating ReGRAF into the baselines introduces minimal additional running time. Furthermore, Tab. 4 and Tab. 5 demonstrate that even with hyperparameters tuned on 10 randomly sampled instances from COCO-20$^i$, ReGRAF effectively improves the baseline's segmentation performance across diverse datasets.

