# OpenReview forum: "ReGRAF: Training free Prompt Refinement via Gradient Flow for Segmentation"
_ICLR.cc/2025/Conference — ICLR 2025 Conference Withdrawn Submission_

### Official Review · Reviewer_snkx · 2024-10-28

**Soundness:** 3
**Presentation:** 3
**Contribution:** 3
**Rating:** 5
**Confidence:** 3

**Summary:**

Despite the integration of SAM into one-shot and few-shot segmentation, which enables auto-prompting through semantic alignment between query and support images, there are limitations in the generated prompts that can degrade the quality of segmentation. This is primarily due to visual inconsistencies between the support and query images.
To address this issue, the authors propose a training-free method called ReGRAF (Refinement via Gradient Flow). ReGRAF refines the prompts by utilizing gradient flow derived from SAM's mask decoder. This approach aims to improve the quality of prompts without the need for additional training.

**Strengths:**

The proposed method serves as a simple and effective training-free refinement technique, offering broad applicability to Auto-Prompting Segmentation Frameworks. Additionally, the authors provide a theoretical guarantee of convergence, further strengthening the credibility and reliability of their approach.

**Weaknesses:**

1.	In table 1 and 2, the incremental improvement resulting from the addition of ReGRAF is not readily apparent. This raises concerns about whether the implementation of ReGRAF will yield a satisfactory increase in performance, considering the trade-off in terms of time and cost.
2.	When considering the accuracy of SAM segmentation, it is often observed that the mask fails to accurately calibrate complex edges. This issue becomes particularly challenging when dealing with cases involving complex edges or thin objects. What happen for this hards cases with complex edges or thin objects by your ReGRAF. It is happy to add more high-accurate segmentation wokrs:
[1]. Transparent Image Layer Diffusion using Latent Transparency
[2]. DiffuMatting: Synthesizing Arbitrary Objects with Matting-level Annotation
[3]. Highly Accurate Dichotomous Image Segmentation

**Questions:**

see in weakness

---

> ### Author Response · Authors · 2024-11-21
>
> Thank you for the insightful comments and suggestions. Your thorough review and valuable feedback have guided us in enhancing both the precision and presentation of our work. In addressing your questions, we carefully conducted additional experiments, which allowed us to further validate and refine our findings. We trust these efforts have strengthened the overall rigor of our study.
>
> - **The effectiveness of ReGRAF?** To better demonstrate the effectiveness of ReGRAF, we presented average mIoU segmentation results of each dataset as well as the oracle results in the appendix D. The oracle results are obtained by selecting the most accurate segmentation masks across the maximum iterations (T=5) based on a comparison with the ground truth, rather than simply choosing the mask from the specific iteration. This approach closely resembles real-world usage of ReGRAF, as it allows selecting the most suitable segmentation result from different iterations.\
>     As time was limited, we first performed the oracle test on PerSAM/HQ-PerSAM. We will update this oracle results for the other baselines (i.e., Matcher, HQ-Matcher) as soon as possible.\
>    Our results demonstrate that while the optimal iteration may vary across samples, ReGRAF consistently refines segmentation masks effectively, yielding significant performance improvements in practical applications. (For example, PerSAM achieves at least a 4.26% improvement in segmentation accuracy, with a maximum improvement of 15.62%.)
>
> - **Additional experiment result for fine-grained segmentation dataset?** To show the performance of ReGRAF on the segmentation of fine-grained datasets, we validated our method on the DIS5K dataset ([3]). We chose DIS5K over the other datasets suggested by the reviewer for the following reasons:
>
>    - The data used in [1] is not publicly available,
>    - We argue that datasets with less distinct separation between foreground and background are better suited for evaluating more challenging cases,
>    - DIS5K is publicly available, features poor separation between foreground and background, and includes class metadata for few-shot guidance.
>
>   We recorded the segmentation results without further hyper-parameter tuning (We applied the same hyperparameter settings for ReGRAF as those used in part segmentation.) Each element from iter1 to iter5, as well as the oracle, represents the mIOU gain.\
>    Although the progression of average mIoU gains suggests challenges in selecting the optimal maximum iterations T, the oracle results demonstrate that our method effectively enhances segmentation quality, even for this difficult task (This table is also attached in Appendix E).
>
>    | Baselines | Baseline | iter 1 | iter 2 | iter 3 | iter 4 | iter 5 | Oracle |
>    |-------------|----------|----------|----------|----------|----------|----------|-----------|
>    | PerSAM-F | 27.47 | 0.00 | **0.24** | **0.33** | **0.02** | -0.24 | **6.33** |
>    | HQ-PerSAM-F | 52.62 | 0.00 | -0.12 | **0.06** | **0.17** | **0.11** | **5.55** |
>    | Matcher | 46.77 | -0.29 | **0.14** | -0.19 | -0.23 | **0.14** | **10.87** |
>    | HQ-Matcher | 58.39 | **0.09** | **0.58** | -0.06 | **0.18** | **0.52** | **10.65** |

---

### Official Review · Reviewer_a5bV · 2024-11-02

**Soundness:** 2
**Presentation:** 3
**Contribution:** 3
**Rating:** 5
**Confidence:** 4

**Summary:**

This paper delves into the domain of prompt segmentation, which is pertinent to the foundational model, SAM. The authors propose a training-free method to refine segmentation results by updating the visual embeddings of query images. They provide a rigorous theoretical analysis to elucidate why the gradient flow through the mask decoder can optimize segmentation results and demonstrate the convergence of their approach. They assert that this design can be readily integrated with other baseline methods for prompt segmentation, which is exemplified in the experimental section.

**Strengths:**

1.This paper provides a substantial theoretical justification for the method, which possesses favorable interpretability.
2.The method can be integrated flexibly with multiple baseline segmentation prompts without the need for training, while incurring minimal additional inference costs.
3.The paper includes a wealth of visualized results.

**Weaknesses:**

The analysis of the experimental results in the article is insufficient, and the improvement achieved is limited. Firstly, the paper only compares their method with  PerSAM-F， Matcher and their HQ_SAM variants, without benchmarking against other methods in the field. The analysis in the related work section is similarly inadequate. Secondly, apart from the comparisons with baseline methods in Tables 1 and 2, there is a lack of analytical experiments and an examination of hyper-parameters. Thirdly, the improvements shown in Tables 1 and 2 are overall modest, with many of the increments less than 0.5%. Given that segmentation task results typically exhibit significant variability, these results provide insufficient evidence of the method's efficacy.

**Questions:**

1、Why does the performance of ReGRAF, when integrated with HQ-Matcher on the COCO-20i dataset, fall short of the results achieved with the standard Matcher?
2. Please explain line 359, "gradient estimation in equation (12) becomes inaccurate with a large number of iterations." Why does an increase in the number of iterations lead to inaccuracies in the results? Theoretically, shouldn't the process gradually converge to a value near the optimal one?
3.In section 4.2.2, the author states that the numerical improvement is not significant, explaining that the segmentation effect is better, but these individual examples are not very persuasive. Perhaps some better evaluation criteria could more comprehensively demonstrate the enhancement brought by the method.

---

> ### Author Response · Authors · 2024-11-20
>
> We sincerely appreciate the insightful feedback. Your thoughtful comments and questions have played a key role in improving the clarity and quality of our work. Through our revisions, we aimed to carefully address your suggestions and further strengthen the presentation of our study.
>
> - **Comparing HQ-Matcher and Matcher results on COCO-20$^i$ dataset?** HQ-SAM was fine-tuned using prompts (visual clues) such as bounding boxes surrounding the true mask and random points within it. This fine-tuning likely introduced a bias toward the labels in the training dataset. Additionally, while Matcher's auto-prompting was specifically tailored for SAM, HQ-Matcher incorporated HQ-SAM into ReGRAF without further tuning to emphasize the _broad applicability_ of our approach. As a result, HQ-SAM’s HQ features may have been derived from incomplete prompts, leading to an out-of-distribution (OOD) scenario that likely constrained HQ-Matcher's improvements over Matcher.
> - **Additional explanation of equation (6), (12)  and Theorem 1.** This is because the density ratio in Equation (12) is derived from the approximation introduced in Equation (6). Specifically, the exact density ratio is represented as $\rho_t / \mu$, where $t$ is intractable. To address this, in Equation (6), we approximated the density ratio as  $\rho_0 / \mu$ (e.g. $\rho_t \approx \rho_0$  ), which is valid for small values of $t$ but not for large $t$. However, this approximation raises questions about the stability of ReGRAF. To resolve this, we proved that, given the near-optimal latent vector of the visual foundation model, ReGRAF converges at an exponential rate (Theorem 1). This proof ensures the stability of our algorithm, liberating it from reliance on the initial approximation.
> - **Better evaluation criteria?**  In the appendix D, we presented average mIoU segmentation results of each dataset as well as the oracle results. The oracle results are obtained by selecting the most accurate segmentation masks across the maximum iterations ($T=5$) based on a comparison with the ground truth, rather than simply choosing the mask from the final iteration. This approach demonstrates practical use of ReGRAF, as it allows selecting the most suitable segmentation result from different iterations.\
> As time was limited, we first performed the oracle test on PerSAM/HQ-PerSAM. We will update this oracle results for the other baselines (i.e., Matcher, HQ-Matcher)  as soon as possible.\
> Our results demonstrate that while the optimal iteration may vary across samples, ReGRAF consistently refines segmentation masks effectively, yielding significant performance improvements in practical applications. (For example, PerSAM achieves at least a 4.26% improvement in segmentation accuracy, with a maximum improvement of 15.62%.)\
> Furthermore, we further validated our method on DIS5K dataset [1]. DIS5K is specifically designed for segmentation tasks where fine-grained objects are difficult to distinguish and require more accurate segmentation masks. We recorded the segmentation results without further hyper-parameter tuning (We applied the same hyperparameter settings for ReGRAF as those used in part segmentation.) Each element from iter1 to iter5, as well as the oracle, represents the mIOU gain.\
> Although the progression of average mIoU gains suggests challenges in selecting the optimal maximum iterations T, the oracle results demonstrate that our method effectively enhances segmentation quality, even for this difficult task (This table is attached in Appendix E).
> | Baselines | Baseline | iter 1 | iter 2 | iter 3 | iter 4 | iter 5 | Oracle |
> |-------------|----------|----------|----------|----------|----------|----------|-----------|
> | PerSAM-F | 27.47 | 0.00 | **0.24** | **0.33** | **0.02** | -0.24 | **6.33** |
> | HQ-PerSAM-F | 52.62 | 0.00 | -0.12 | **0.06** | **0.17** | **0.11** | **5.55** |
> | Matcher | 46.77 | -0.29 | **0.14** | -0.19 | -0.23 | **0.14** | **10.87** |
> | HQ-Matcher | 58.39 | **0.09** | **0.58** | -0.06 | **0.18** | **0.52** | **10.65** |
>
> [1] Qin, X., Dai, H., Hu, X., Fan, D. P., Shao, L., & Van Gool, L. (2022, October). Highly accurate dichotomous image segmentation. In European Conference on Computer Vision (pp. 38-56). Cham: Springer Nature Switzerland.

---

### Official Review · Reviewer_wjwX · 2024-11-02

**Soundness:** 2
**Presentation:** 3
**Contribution:** 2
**Rating:** 5
**Confidence:** 4

**Summary:**

Though SAM has achieved impressive generalization ability. It is not directly tailored to be specific to different tasks. Recently, one-shot and few-shot methods such as PerSAM and Matcher have been proposed to generate the auto prompting via query and supporting set. However, the auto-prompt generation process is far from perfect, missing prompts or error prompts can appear. As a result, the resulting mask is not aligned with the support set well. This paper proposed to leveraging additional cues to iteratively sample prompts for improving one/few-shot segmentation performance. As the ground truth masks are not available for correcting the prompts, the author proposed to leverage gradient flow from the decoder to improve the prompt sampling, and therefore, the final segmentation results.

**Strengths:**

Leveraging the guidance from the decoder to guide the prompt sampling for interactive segmentation is novel from my perspective. It alleviates the need for external human supervision and makes use of the interactive correction ability of the interactive segmentation foundation model. Specifically, the author computes the gradient flow from the decoder to align the distribution of the query and support embeddings. Afterward, the updated query embedding is used to compute the similarity map for a new round of interactive segmentation.

The proposed method has been validated on both the part segmentation dataset and semantic segmentation, which demonstrates its broad applicability.

A theoretical analysis has been provided to show the proposed method's convergence and convergence rate.

The presentation of the paper is well-structured and the equations are clearly displayed.

**Weaknesses:**

The performance improvement is somehow marginal for both the semantic segmentation and part segmentation, across dataset and base method. From the qualitative examples, it seems that the refinement process is quite effective in aligning different semantic hieratical levels of the target and support mask. For example, in figure 5, some object-level base predictions are correctly adjusted to part masks with the proposed methods. In figure 4, some missing components are correctly saved back. Does the proposed method aim to resolve the semantic hieratical ambiguity or mainly the details correction? If it is the first part, then it is unclear to me how is this achieved. As it is not well explained in the method part or the motivation part.

Besides, the examples displayed achieve significant improvement. However, quantitative evaluation shows the opposite. Are there some examples becoming worse after refinement? It would be nice to show some scatter plots of the sample-wise IoU before and after refinement so that we have a better idea of the advantages and the limitations of the proposed method.

I miss the ablation on the number of refinement times.

**Questions:**

See the weakness. I am willing to raise the score if the author can solve my concerns

---

> ### Author Response · Authors · 2024-11-20
>
> We are grateful for the detailed and valuable feedback from the reviewers. Your insightful questions and suggestions have encouraged us to refine our work and strengthen its contributions. We trust that our revisions adequately address your concerns and further enhance the quality of this study.
>
> - **More clear motivation and objective of ReGRAF?** Our method was not meticulously designed to address the issues that you suggested, but to  exploit the gradient flow to draw out the hidden potential of the Visual Foundation Model without incurring additional expenses in terms of training, dataset, parameter, etc. ReGRAF's value in segmentation refinement lies in its ability to improve segmentation quality as a training-free method. It requires minimal additional computation time (as highlighted in Table 3 of Section 4.2.3) and only minor code modifications for implementation. This makes ReGRAF broadly applicable to auto-prompting frameworks.
> - **Failure cases?** Yes, in some cases, performance declines after refinement. Our method is based on the strong assumption that the Vision Foundation Model (VFM) obtains latent vectors for query images that are already near-optimal. However, in real-world applications (E.g., out-of-distribution settings) the VFM sometimes fails to capture a near-optimal latent vector. In these scenarios, refinement may lead to poorer segmentation results (Please refer to the appendix C for the failure cases.).
> - **Sensitivity of iteration times?** We undertook a sensitivity analysis on the step size $\eta$ and the iterations $T$ in the appendix F. We randomly sampled 100 examples from the set-aside dataset (derived from the COCO-20$^i$ training dataset) and we evaluated the mIoU over iterations, testing up to 30 iterations with step sizes of $10^{-2}, 10^{-3}, 10^{-4}$ and $10^{-5}$). Figure 16 in the appendix F shows a moving average with a window size of 2 to smooth fluctuations and better reveal overall trends. \
> The results emphasize the critical role of the step size ($\eta$) in segmentation performance and stability. $\eta=10^{-3}$ consistently delivered the best results, balancing rapid convergence and long-term stability. In contrast, $\eta=10^{-2}$ improved quickly but degraded over time, while smaller values ($\eta=10^{-2}, \eta=10^{-5}$) showed slower convergence and higher variability.\
> This underscores the importance of tuning the step size, however ReGRAF can be easily tuned and applicable to various scenarios. As shown in Table 3, incorporating ReGRAF into the baselines introduces minimal additional running time. Furthermore, Table 4 and Table 5 demonstrate that even with hyperparameters tuned on 10 randomly sampled instances from COCO-20$^i$, ReGRAF effectively improves the baseline's segmentation performance across diverse datasets.

---

### Official Review · Reviewer_XVBq · 2024-11-03

**Soundness:** 3
**Presentation:** 3
**Contribution:** 2
**Rating:** 6
**Confidence:** 4

**Summary:**

This paper proposes  ReGRAF (Refinement via Gradient Flow), which is a training free module that can be added to auto-prompting methods (such as PerSAM or Matcher) for promotable segmentation models (such as SAM). ReGRAF relies on using gradient flow, logits from mask decoder and query embeddings to update embeddings of the query, to best match that of support image. ReGRAF is an iterative method that iteratively updates embeddings with no need to learnable parameters.

Authors tested the proposed method on 2 tasks of semantic segmentation and part segmentation on total of 5 different datasets and a combination of 7 baselines stemmed from variations on 2 major baseline methods (PerSAM or Matcher)

**Strengths:**

- Although iterative, the method's running time is marginal compares to baselines (Table 3)
- That paper is easy to follow and formulations clearly explain the proposed method.

**Weaknesses:**

- Although the proposed method provides consistent improvement on top of baselines, the improvements are mostly very marginal (less than 0.5% in majority of times and less than 1% in almost every experiment except for 2 on a specific dataset (Tables 1 and 2).
- The visual results in the paper mostly focused on cases where the proposed method improved baselines and there is a lack of showing failure cases. This causes a lack of clarity on weaknesses of the method and a fair understanding of performance.
- The proposed method seem to rely on a number of hyper parameters (Iterations T, entropy regularization factor, and step size). The paper does not provide any experiments to show sensitivity of the method to these parameters and their impact on performance.

**Questions:**

- Given the marginal improvements this begs the question of how much time should one spend tuning these parameters for new cases and how sensitive the method will be to them?
- Authors mentioned "s equation (6) implies that the gradient estimation in equation (12) becomes inaccurate with a large number of iterations." (on page 7 section Hyperparameter setting of ReGRAF). This was not clear to me. How in accurate and why?
- Table 1 shows a wide range of performance improvements depending on the baseline and dataset. Would be great is authors can share some insight for when their method could have more significant improvements vs marginal. Was there any trends in the results?
- What are some cases that the method will fail and provide worse results than baseline?

---

> ### Author Response · Authors · 2024-11-20
>
> We deeply appreciate the reviewers for their careful evaluation and valuable feedback. Your constructive suggestions and thoughtful questions have significantly enhanced the clarity and quality of our work. We sincerely hope our detailed responses and improvements meet your expectations and contribute positively to your overall assessment.
>
> - **Sensitivity of hyperparameters?** Since we assumed that the visual foundation model yields a near-optimal segmentation, we did not take much time on tuning the hyper-parameters, using a single set-a-side dataset for the entire experimental settings (10 random samples from COCO-20i set-aside dataset). Depending on the purpose of refinement, we employed different step size . Specifically, we used $10^{-3}$ when understanding overall visual concept would be important, e.g., semantic segmentation, and $10^{-4}$ (which is relatively small) when the segmentation would require capturing fine details in a scene, e.g., part segmentation.
> - **Additional explanation of equation (6), (12)  and Theorem 1.** This is because the density ratio in Equation (12) is derived from the approximation introduced in Equation (6). Specifically, the exact density ratio is represented as $\rho_t / \mu$, where $t$ is intractable. To address this, in Equation (6), we approximated the density ratio as  $\rho_0 / \mu$ (e.g. $\rho_t \approx \rho_0$  ), which is valid for small values of $t$ but not for large $t$. However, this approximation raises questions about the stability of ReGRAF. To resolve this, we proved that, given the near-optimal latent vector of the visual foundation model, ReGRAF converges at an exponential rate (Theorem 1). This proof ensures the stability of our algorithm, liberating it from reliance on the initial approximation.
> - **The trends of results?** For segmenting small objects or fine details, using a smaller step size is advantageous to prevent abrupt changes in the prompt. However, when the visual semantics between the reference image and the target image differ significantly (4th rows in the appendix C.), prompt refinement becomes challenging, potentially leading to performance degradation. \
> In addition, when the visual instructions in the reference images are difficult to interpret— such as when distinguishing between specific parts of a bicycle is challenging, or when the reference depicts a general tray while the segmentation target is the tray's edge—ReGRAF struggles to refine the prompts effectively (2nd, 3rd, and 5th rows in the appendix C).

---

### Author Response · Authors · 2024-11-20

Dear Reviewers,

We sincerely thank the reviewers for their valuable feedback and thoughtful suggestions. Your comments significantly help us improve the clarity and quality of our work. To summarize the strengths of our method that the reviewers appreciated, they are as follows:

- Theoretical robustness
   - “A theoretical analysis has been provided to show the proposed method's convergence and convergence rate.” (Reviewer wjwX)
    - “This paper provides a substantial theoretical justification for the method, which possesses favorable interpretability.” (Reviewer a5bV)
    - “The authors provide a theoretical guarantee of convergence, further strengthening the credibility and reliability of their approach.” (Reviewer snkx)
- Effectiveness
   - “Although iterative, the method's running time is marginal compares to baselines” (Reviewer XVBq)
   - “The method can be integrated flexibly with multiple baseline segmentation prompts without the need for training, while incurring minimal additional inference costs” (Reviewer a5bV)
   - “The proposed method has been validated on both the part segmentation dataset and semantic segmentation, which demonstrates its broad applicability.” (Reviewer wjwX)
   - “The proposed method serves as a simple and effective training-free refinement technique, offering broad applicability to Auto-Prompting Segmentation Frameworks” (Reviewer snkx)
- Clear presentation
   - “That paper is easy to follow and formulations clearly explain the proposed method.” (Reviewer XVBq)
   - “The presentation of the paper is well-structured and the equations are clearly displayed.” (Reviewer wjwX)
   - “The paper includes a wealth of visualized results.” (Reviewer a5bV)

We deeply appreciate again for the positive feedback provided by the reviewers. At the same time, we value the insightful comments regarding limitations or unclear aspects of our study.

Common questions include:
1. Justification for the marginal improvement in the evaluation metric.
2. Sensitivity analysis of the hyperparameters (e.g., the number of iterations and step size).
3. Clarification regarding Theorem 1.

To address these points, we have revised a part of the manuscript related to the experiments, and added supplementary materials (Appendix C~F). We highlighted the revised or newly introduced contents in red for your reference.

We outline our responses to the common questions in the following.

1. **Additional discussion on ReGRAF's performance.**

   1.1 We included failure cases of ReGRAF in the appendix C to illustrate the usual cases that our method fails to refine prompts.

   1.2 We reported oracle results of ReGRAF to better validate the refinement performance of our method in Appendix D and E.
   - Here, the oracle test chooses the most accurate segmentation masks across the maximum iterations ($T=5$) based on a comparison with the ground truth, rather than simply choosing the mask from the specific iteration. This approach closely resembles the practical usage of ReGRAF, as it allows selecting the most suitable segmentation result from different iterations.

   1.3 We provided a more detailed explanation of the process through which Tables 1 and 2 were obtained (in Section 4.1 Experiment setting - dataset and evaluation).
   - In specific, we averaged mIoU of each dataset and also listed the mIoU gain of ReGRAF from baselines across iterations (within $T=5$). The best average mIoU per dataset within the total iterations of both semantic and part segmentation are reported in Table 1 and 2 in the main paper. **We genuinely apologize for any confusion and sincerely hope you can understand.**

2. **Sensitivity Analysis.**

   We undertook the sensitivity analysis on the step size $\eta$ and iterations $T$ in the appendix F to address concerns regarding the hyperparameter selection for ReGRAF.

3. **Clarification regarding Theorem 1**

   Regarding Equation (12) becoming inaccurate with a large number of iterations, this is because of the approximation in Equation (6). However, Theorem 1 shows that ReGRAF will stably converge within a small number of iterations, before the approximation in Equation (12) becomes too inaccurate.

---

> ### Author Response · Authors · 2024-11-23
>
> We presented the complete updated *oracle* results in Appendix D.

---

> ### Author Response · Authors · 2024-11-27
>
> Dear Reviewers,
>
> We sincerely hope that our rebuttal has addressed your concerns and clarified our work based on your feedback.
>
> We greatly value your comments, and if you have any additional questions or need further clarification, please let us know before the PDF revision period ends (Nov. 27 23:59 (AoE)). We look forward to contributing to the ICLR and the broader ML/CV community through the publication of this paper.

---

### Author Response · Authors · 2024-12-01
**Discussion**

Dear Reviewers,

We would be happy to be engaged in discussions to clear any doubts you may have.
We have presented our rebuttal, so please let us know if you have any questions to reconsider your evaluations.

---

### Note · Authors · 2025-03-28

I have read and agree with the venue's withdrawal policy on behalf of myself and my co-authors.

---

### Meta-Review · Area_Chair_boMQ · 2024-12-22

**Metareview:**

This work introduces ReGRAF (Refinement via GRAdient Flow), which is a training-free method that refines prompts through gradient flow derived from SAM's mask decoder. ReGRAF seamlessly integrates into SAM-based auto-prompting frameworks and is theoretically proven to refine segmentation masks with high efficiency and precision.

**Additional Comments On Reviewer Discussion:**

This work has four reviewers. Three reviewers are negative with scores of 5, 5, and 5.  The other review is positive to accept this work with a score 0f 6. In this regard, this work can not be accepted.

---

### Decision · Program_Chairs · 2025-01-22

Reject